# Learning Protein Structure-Function Relationships through Knowledge-guided Representation Decomposition

**Mingqing Wang** [1 2]  **Zhiwei Nie** [2 3]  **Athanasios V. Vasilakos** [4]  **Yonghong He** [1]  **Zhixiang Ren** [2 5]

## Abstract

Proteins encode diverse functions within complex three-dimensional structures, yet most deep learning representations remain highly entangled, obscuring the biophysical signals that underlie function. Here we introduce ProtDiS, a knowledge-guided framework that decomposes pretrained protein micro-environment embeddings into biologically grounded and task-relevant dimensions. Inspired by the information bottleneck principle, ProtDiS learns representations that balance informativeness and compression, yielding structural features that are more specific, independent, and information-efficient, and achieving consistent improvements across twelve downstream tasks, with the largest gains under structure-based splits. Protein- and residue-level analyses further show that ProtDiS differentiates proteins with similar folds but divergent functions and captures fine-grained biophysical signals critical. These findings suggest that knowledge-guided decomposition provides a general and interpretable approach for structuring latent spaces in protein structural modeling. The source code and implementation details are publicly available at https://github.com/AI-HPC-Research-Team/ProtDiS.

## 1. Introduction

A growing challenge in protein modeling is to understand and constrain the latent spaces learned by deep neural networks, so that structural information can be organized, manipulated, and interpreted in biologically meaningful ways. Although breakthroughs in structure prediction (Abramson et al., 2024; Baek et al., 2021), generative design (Hayes et al., 2025), and inverse folding (Dauparas et al., 2022) have enabled powerful models, their internal representations remain entangled and difficult to relate to concrete biophysical principles. Recent efforts attempt to impose structure on latent spaces by introducing external knowledge or constraints, including multimodal alignment (Su et al., 2025a), expert-guided post-training (Duan et al., 2025), physical regularization (Gelman et al., 2025), sparse concept discovery (Simon & Zou, 2025), and biologically informed contrastive learning (Peng et al., 2025). However, most learned embeddings still intermix geometric, physicochemical, and topological signals, limiting mechanistic interpretability and generalization across folds.

We approach this problem from the perspective that protein function depends not on the full high-dimensional structural embedding, but on a small set of semantically meaningful and mechanistically grounded properties of local micro-environments, such as secondary structure, packing density, flexibility, or geometric curvature. These properties act as intermediate variables that mediate the relationship between structure and function, and are more stable than global structural similarity under perturbations. From this viewpoint, the goal is to transform entangled structural representations into disentangled, knowledge-aligned factors that selectively preserve functionally relevant information.

To this end, we propose ProtDiS, a knowledge-guided representation factorization framework that decomposes a pretrained structural embedding into multiple interpretable knowledge channels (Figure 1). Each channel is explicitly aligned with a predefined biophysical or geometric attribute, enabling independent manipulation, selective reuse, and improved interpretability in downstream tasks. Rather than enforcing strict statistical independence—which is unrealistic given correlated biophysical properties—ProtDiS adopts a redundancy-reduction principle that encourages functional disentanglement while preserving complementary information.

The proposed framework can be interpreted through an in-

[1]Shenzhen International Graduate School, Tsinghua University, Shenzhen, China [2]Pengcheng Laboratory, Shenzhen, China [3]School of Electronic and Computer Engineering, Peking University, Shenzhen, China [4]CAIR, University of Agder, Norway [5]Shanghai Smart Logic Technology Co. Ltd., Shanghai, China. Correspondence to: Zhixiang Ren <jason.zhixiang.ren@outlook.com>, Athanasios V. Vasilakos <Thanos.vasilakos@uia.no>.

*Proceedings of the $43^{rd}$ International Conference on Machine Learning*, Seoul, South Korea. PMLR 306, 2026. Copyright 2026 by the author(s).

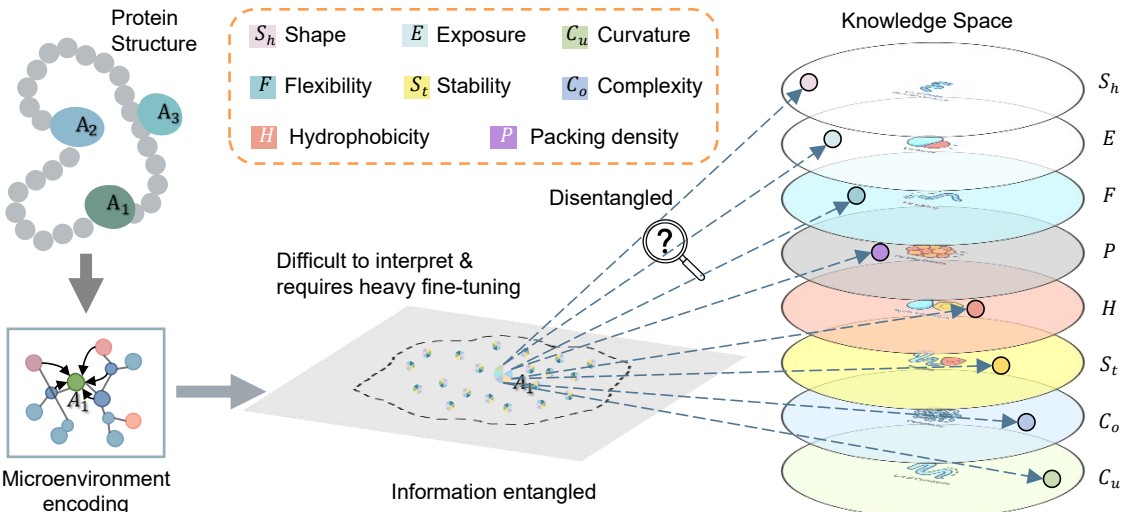

*Figure 1.* Overview of the proposed ProtDiS framework. We disentangle entangled structural representations into eight semantically grounded knowledge spaces, each corresponding to a specific local structural or physicochemical property.

formation bottleneck lens. Each knowledge channel learns a compressed representation that maximizes mutual information with its corresponding knowledge signal while minimizing redundant information shared with other channels. To avoid lossy or degenerate factorization, ProtDiS maintains an additional residual channel that captures structural information not explained by the predefined knowledge axes. The original structural embedding is reconstructed from the set of knowledge channels together with this residual, ensuring information completeness while keeping each knowledge channel minimally sufficient for its target property.

From a causal and invariance perspective, the learned knowledge channels approximate mechanism-level variables that govern protein function. By explicitly isolating such variables, ProtDiS learns representations that are less sensitive to superficial fold similarity and more aligned with functionally relevant mechanisms, explaining its improved performance under structure-based splits and in distinguishing proteins with similar folds but divergent functions.

## 2. Related Works

### 2.1. Protein Structural Modeling

Protein structural modeling seeks to encode three-dimensional conformations into representations that support function prediction and transfer learning. Existing approaches span a range of architectures, including graph neural networks with geometric inductive biases (Jing et al., 2021; Gligorijević et al., 2021) and transformer-based models with equivariant attention mechanisms (Frank et al.,

2024), which together establish effective paradigms for learning structure-aware representations. More recently, large-scale self-supervised pretraining has become central to protein structural representation learning, with models such as GearNet (Zhang et al., 2023) and Pythia (Sun et al., 2025) demonstrating strong transferability across tasks. An increasingly influential direction focuses on pretrained protein micro-environment representations, which encode localized structural neighborhoods around residues or functional sites. Approaches such as Foldseek (Van Kempen et al., 2024) and ESM3 (Hayes et al., 2025) show that micro-environment embeddings are highly expressive and effective for large-scale structural search (Van Kempen et al., 2024), clustering (Barrio-Hernandez et al., 2023), structure-aware protein language modeling (Su et al., 2024; 2025b), and catalytic function prediction (Derry et al., 2025). Despite their success, these representations typically entangle geometric, physicochemical, and topological signals within a single latent space, limiting interpretability and controllable generalization. Our work builds upon pretrained micro-environment representations and explicitly decomposes them into disentangled, knowledge-aligned components to better capture structure–function relationships.

### 2.2. Disentangled Representation Learning

Disentangled representation learning aims to factorize latent spaces into components corresponding to distinct generative factors, with a prominent line of work grounded in information-theoretic principles. Representative methods such as InfoVAE (Zhao et al., 2019), FactorVAE (Kim &

Mnih, 2018), and $\beta$-TCVAE (Chen et al., 2018) introduce mutual-information and total-correlation–based objectives to reduce dependency among latent factors, while supervised extensions such as DisenIB (Pan et al., 2021) and the infomax–bottleneck (IMB) framework (Yang et al., 2022) further align disentanglement with task relevance and knowledge abstraction.

Recent studies explore disentanglement in protein representation learning. Sparse autoencoder–based analyses of protein language model embeddings (Adams et al., 2025; Simon & Zou, 2025) identify interpretable features associated with structural motifs, binding sites, and functional domains, and Pantolini et al. (2025) decompose protein language model outputs into a small set of structural codewords using contrastive learning and deep clustering. While effective for post hoc interpretability, these approaches typically operate on sequence-level embeddings and lack explicit information-theoretic constraints tied to structural or physical knowledge. In contrast, our work formulates disentanglement from first principles using information-theoretic objectives combined with knowledge supervision, explicitly decomposing pretrained protein micro-environment representations into structure-aligned factors and enabling the study of causal links between structural knowledge components and protein function.

# 3. Methods

## 3.1. Problem Setup and Notation

Given a protein structure, we first obtain a high-dimensional structural representation using a pretrained structural encoder (e.g., ESM-3). We denote this representation as a random variable

$$\mathbf{s} \in \mathbb{R}^d, \tag{1}$$

which encodes rich geometric and physicochemical information but is typically highly entangled.

In addition to the structural embedding, we consider a set of predefined, biologically grounded knowledge variables

$$\mathcal{Y} = \{Y_1, Y_2, \ldots, Y_K\}, \tag{2}$$

where each $Y_k$ corresponds to a specific local structural or physicochemical property, such as secondary structure, solvent accessibility, flexibility, local packing, or geometric curvature. These variables are computed directly from protein structures and serve as explicit semantic supervision.

Our objective is to transform the entangled structural embedding $\mathbf{s}$ into a set of latent representations consisting of: (i) a collection of knowledge-specific channels

$$\mathcal{Z} = \{Z_1, Z_2, \ldots, Z_K\}, \tag{3}$$

and (ii) an additional residual channel

$$Z_c, \tag{4}$$

such that each $Z_k$ selectively captures information relevant to $Y_k$, while the joint representation $(Z_1, \ldots, Z_K, Z_c)$ preserves the full information content of the original embedding $\mathbf{s}$.

## 3.2. Knowledge-Guided Representation Factorization via Information Bottleneck

We propose a knowledge-guided representation factorization framework that decomposes the structural embedding $\mathbf{s}$ into multiple parallel latent channels. Each knowledge channel is defined by a parameterized encoder

$$Z_k = f_k(\mathbf{s}), \quad k = 1, \ldots, K, \tag{5}$$

while the residual channel is obtained via a separate encoder

$$Z_c = f_c(\mathbf{s}). \tag{6}$$

Our framework draws inspiration from the Information Bottleneck (IB) principle. Theoretically, we aim to learn representations $Z_k$ that maximize predictive power for $Y_k$ while compressing irrelevant information from $\mathbf{s}$, formulated as:

$$\min_{Z_k} \; I(Z_k; \mathbf{s}) - \beta_k I(Z_k; Y_k). \tag{7}$$

Under this objective, each knowledge channel approximates a minimal sufficient statistic of the structural embedding with respect to $Y_k$. Unlike unsupervised disentanglement, the factorization is explicitly anchored to semantically meaningful variables, ensuring interpretability and functional relevance.

In addition to preserving structural information, the residual channel $Z_c$ is explicitly constrained to exclude information related to the predefined knowledge variables. Formally, the objective of the residual channel can be expressed as

$$\max_{Z_c} \; I(Z_c; \mathbf{s}) \; - \; \gamma \sum_{k=1}^{K} I(Z_c; Y_k), \tag{8}$$

where $\gamma$ controls the strength of knowledge invariance. This formulation encourages $Z_c$ to capture complementary structural information while remaining orthogonal, in an information-theoretic sense, to all knowledge-specific channels.

Together, the knowledge channels and the residual channel form a complete and structured decomposition of the original representation:

$$(Z_1, \ldots, Z_K, Z_c) \; \text{jointly sufficient for } \mathbf{s}. \tag{9}$$

### 3.3. Regularization and Optimization Objectives

The information bottleneck formulation in Section 3.2 is not optimized directly. Instead, we derive a practical training objective by decomposing it into several tractable loss components.

**Knowledge Supervision Loss.** To maximize the mutual information between each knowledge channel $Z_k$ and its corresponding knowledge variable $Y_k$, we introduce a supervised prediction head $h_k(\cdot)$. The knowledge supervision loss is defined as

$$\mathcal{L}_{\mathrm{kn}}^{(k)} = \mathbb{E}_{(\mathbf{s}, y_k)} \left[ \ell\big(h_k(Z_k), y_k\big) \right], \quad (10)$$

where $\ell(\cdot)$ denotes a task-specific loss (e.g., cross-entropy or mean squared error).

Minimizing $\mathcal{L}_{\mathrm{kn}}^{(k)}$ maximizes a variational lower bound of the mutual information $I(Z_k; Y_k)$, thereby enforcing knowledge alignment for each channel.

**Bottleneck Regularization Loss.** To encourage compact and non-redundant representations in each knowledge channel, we impose a KL-divergence–based bottleneck regularization. Specifically, each knowledge embedding $Z_k$ is encouraged to match a factorized standard Gaussian prior, which constrains the information capacity of the representation and provides an upper bound on the mutual information between $Z_k$ and the original structural embedding $\mathbf{s}$.

Formally, let $q(Z_k)$ denote the empirical distribution of the $k$-th knowledge embedding induced by a mini-batch. The bottleneck loss is defined as:

$$\mathcal{L}_{\mathrm{KL}} = \sum_{k=1}^{K} \mathrm{KL}\big(q(Z_k) \parallel \mathcal{N}(\mathbf{0}, \mathbf{I})\big), \quad (11)$$

which admits a closed-form expression under a diagonal Gaussian approximation:

$$\mathcal{L}_{\mathrm{KL}} = \frac{1}{2} \sum_{k=1}^{K} \mathbb{E}_d \left[ \mu_{k,d}^2 + \sigma_{k,d}^2 - \log \sigma_{k,d}^2 - 1 \right], \quad (12)$$

where $\mu_{k,d}$ and $\sigma_{k,d}^2$ denote the batch-level mean and variance of the $d$-th dimension of $Z_k$, respectively.

Minimizing $\mathcal{L}_{\mathrm{KL}}$ limits the information capacity of each knowledge channel by regularizing the aggregate latent distribution. This formulation is inspired by the information bottleneck principle, but is implemented as a deterministic, batch-level distributional regularization rather than a per-instance variational bottleneck.

**Reconstruction Loss.** To preserve information from the original structural embedding, we introduce a reconstruction

head $r(\cdot)$ implemented as a two-layer MLP:

$$\hat{\mathbf{s}} = r(Z_c). \quad (13)$$

We enforce reconstruction consistency using an $\ell_1$ loss:

$$\mathcal{L}_{\mathrm{rec}} = \frac{1}{N} \sum_{n=1}^{N} \left\| \hat{\mathbf{s}}_n - \mathbf{s}_n \right\|_1. \quad (14)$$

From an information-theoretic perspective, this objective minimizes the conditional entropy $H(\mathbf{s} \mid Z_c)$. Since $H(\mathbf{s})$ is fixed, minimizing $\mathcal{L}_{\mathrm{rec}}$ maximizes a variational lower bound of the mutual information $I(Z_c; \mathbf{s})$. As a result, the residual channel retains complementary information necessary for completeness.

**Adversarial Knowledge Removal Loss.** To minimize the mutual information between the residual channel $Z_c$ and each knowledge variable $Y_k$, we employ an adversarial training strategy. For each knowledge variable, a discriminator $d_k(\cdot)$ is trained to predict $Y_k$ from $Z_c$, while the residual encoder is trained to prevent such prediction via gradient reversal. The adversarial loss is defined as

$$\mathcal{L}_{\mathrm{adv}} = \sum_{k=1}^{K} \mathbb{E}_{(\mathbf{s}, y_k)} \left[ \ell\big(d_k(\mathcal{R}_\lambda(Z_c)), y_k\big) \right]. \quad (15)$$

where $\mathcal{R}_\lambda(\cdot)$ denotes the gradient reversal operator. Optimizing this loss in an adversarial manner minimizes an upper bound on the mutual information $I(Z_c; Y_k)$, thereby enforcing invariance of the residual channel to all predefined knowledge variables.

**Redundancy Reduction Loss.** While the bottleneck regularization constrains the information capacity of each knowledge channel individually, it does not explicitly prevent different channels from encoding overlapping information. To further encourage complementary and disentangled representations, we introduce a redundancy reduction loss that operates across knowledge channels.

Given a set of knowledge embeddings $\{Z_1, \ldots, Z_K\}$ with $Z_k \in \mathbb{R}^{N \times D}$, we first compute the per-dimension batch statistics of each embedding. Specifically, for each channel $k$ and dimension $d$, we estimate the standard deviation $\mathrm{std}(Z_k^{(d)})$ across the batch. A variance regularization term is applied to the *pre-normalized* embeddings to prevent degenerate or collapsed representations:

$$\mathcal{L}_{\mathrm{var}} = \sum_{k=1}^{K} \mathbb{E}_d \big[ (\mathrm{std}(Z_k^{(d)}) - 1)^2 \big]. \quad (16)$$

Using the same batch statistics, each embedding is then normalized by removing the batch mean and scaling by

the corresponding per-dimension standard deviation. These normalized embeddings are used exclusively for the cross-channel redundancy reduction. For each unordered pair of distinct knowledge channels $(i, j)$, we compute their cross-correlation matrix over the batch:

$$C_{ij} = \frac{1}{N} \tilde{Z}_i^\top \tilde{Z}_j, \quad i \neq j, \tag{17}$$

where $\tilde{Z}_k$ denotes the normalized embedding of channel $k$. To explicitly reduce linear redundancy between channels, we penalize the squared Frobenius norm of these cross-correlation matrices:

$$\mathcal{L}_{\text{cov}} = \frac{1}{|\mathcal{P}|} \sum_{(i,j) \in \mathcal{P}} \|C_{ij}\|_F^2, \tag{18}$$

where $\mathcal{P}$ denotes the set of all unordered pairs of knowledge channels.

This decorrelation objective encourages different knowledge channels to capture complementary information by reducing second-order statistical dependencies, while still allowing nonlinear biological relationships to be preserved. The final redundancy reduction objective is defined as:

$$\mathcal{L}_{\text{red}} = \lambda_{\text{var}} \mathcal{L}_{\text{var}} + \lambda_{\text{cov}} \mathcal{L}_{\text{cov}}. \tag{19}$$

**Final Objective.** The full training objective integrates knowledge supervision, information bottleneck regularization, redundancy reduction, and information preservation. The overall loss is defined as:

$$\mathcal{L}_{\text{total}} = \mathcal{L}_{\text{sup}} + \lambda_{\text{KL}} \mathcal{L}_{\text{KL}} + \lambda_{\text{red}} \mathcal{L}_{\text{red}} + \lambda_{\text{rec}} \mathcal{L}_{\text{rec}} + \lambda_{\text{adv}} \mathcal{L}_{\text{adv}}. \tag{20}$$

where each term plays a complementary role in enforcing disentangled and interpretable representations.

## 4. Experiments

### 4.1. Representation Decoupling Pre-training

In this study, we employ the ESM-3 structural tokenizer as the base encoder for protein micro-environments, leveraging its powerful ability to embed local structure information. However, the ProtDiS framework is model-agnostic and can be readily extended to other structural encoders or multimodal protein language models without architectural modification.

Due to computational resource constraints, we focus on eight core knowledge dimensions, each associated with a quantitative label used as supervision. These include packing density, local complexity, curvature, shape, exposure, flexibility, stability, and hydrophobicity. Although the present study focuses on these eight dimensions, the Prot-DiS architecture naturally generalizes to a larger number

of knowledge components. To accommodate knowledge aspects not explicitly modeled, we introduce an auxiliary residual vector that captures unassigned structural variance. Details of the computation procedures for all quantitative knowledge labels are provided in Section A.2.2.

For pretraining, we curated a large-scale dataset comprising 100,000 protein structures, including high-quality structures sampled from the Protein Data Bank (PDB) and high-confidence chains from the AlphaFoldDB (Fleming et al., 2025). These datasets together provide diverse and reliable coverage of protein folds and local micro-environment configurations.

### 4.2. Feature Analysis

We conduct feature-level analyses to evaluate whether the proposed decomposition produces knowledge channels that satisfy the desired properties of *specificity*, *independence*, and *completeness*. All experiments are performed on 20,000 residue-centered micro-environment embeddings using frozen representations, isolating the effect of representation factorization from downstream model capacity.

**Knowledge-Specificity Analysis.** We assess whether each knowledge channel selectively captures its targeted structural attribute by computing the mutual information (MI) between each knowledge label $Y_k$ and all latent representations, including the eight knowledge channels $Z_1, \ldots, Z_8$ and the residual channel $Z_c$. To enable comparison, we report a normalized MI score relative to the original structural embedding $\mathbf{s}$, $I(Z; Y_k)/I(\mathbf{s}; Y_k)$, computed with the same empirical estimator.

As shown in Figure 2 (left), the resulting heatmap exhibits a pronounced diagonal structure: each knowledge channel attains a high relative MI score primarily for its corresponding label, while non-matching channel–label pairs yield substantially lower values. This pattern indicates that the information relevant to each knowledge label is more readily accessible from its designated channel than from other representations. In contrast, the residual channel consistently shows low MI estimates with all knowledge labels, suggesting that explicit knowledge semantics have been effectively separated from the residual representation. Low-dimensional visualizations further confirm that knowledge channels separate samples only along their targeted attributes, while remaining entangled with respect to unrelated ones (Figure 2, right).

**Independence Across Knowledge Channels.** To evaluate redundancy across channels, we measure pairwise statistical dependence among the original embedding, the residual channel, and all knowledge channels using the Distance Correlation Coefficient (DCC). As shown in Figure 3a, cor-

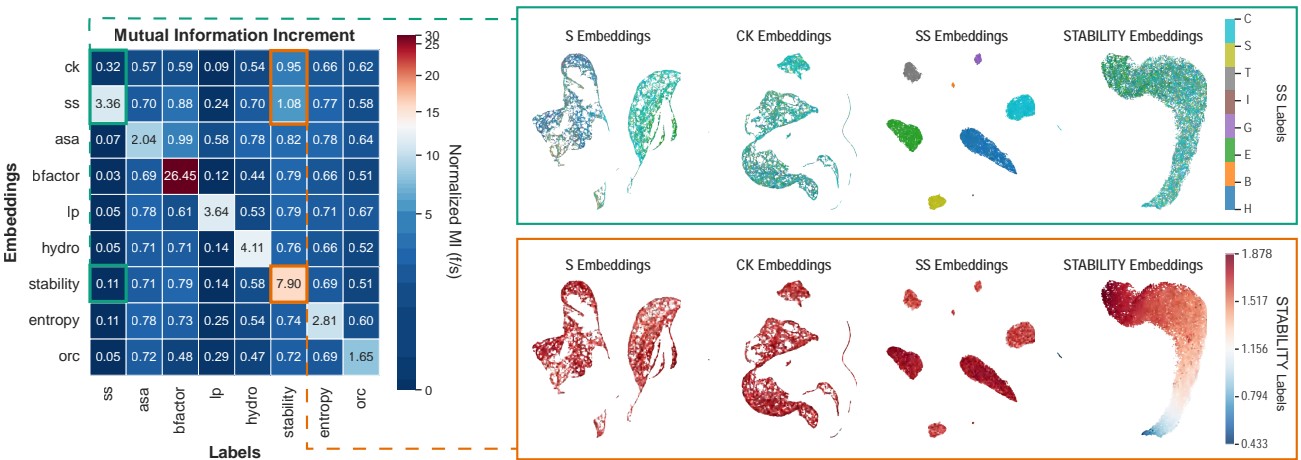

*Figure 2.* Knowledge-specific representation analysis. Left: mutual information gain heatmap comparing each knowledge embedding against the original structural embedding (s-embedding) across different knowledge dimensions. Significant improvements are observed only when the embedding matches its corresponding knowledge channel. Right: distribution visualizations of four types of embeddings under secondary structure and stability labels. Knowledge channels clearly separate their targeted labels, while remaining mixed for unrelated properties, demonstrating strong feature specificity and disentanglement.

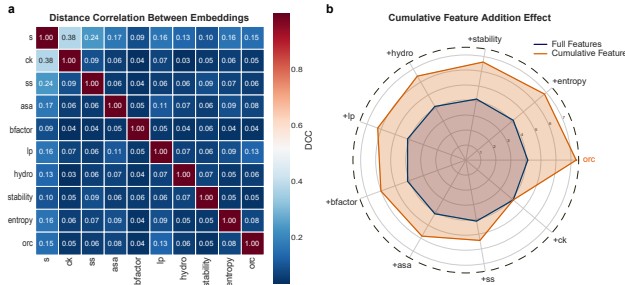

*Figure 3.* Independence and completeness analysis of the knowledge channels. (a) Distance correlation coefficient (DCC) based independence test. Pairwise correlations between different knowledge embeddings are consistently low, indicating effective disentanglement. (b) Progressive reconstruction from knowledge channels visualized as a radar plot. Reconstruction starts from the Ollivier–Ricci curvature (ORC) channel and proceeds counterclockwise by incrementally adding additional knowledge channels.

relations between different knowledge channels remain consistently low, indicating weak cross-channel dependence. At the same time, each knowledge channel retains moderate correlation with the original structural embedding, suggesting that meaningful structural information is preserved rather than discarded.

**Completeness via Progressive Reconstruction.** We further assess whether the set of knowledge channels collectively preserves the full information content of the original

embedding by progressively reconstructing the structural representation from subsets of channels. A lightweight reconstruction network is trained using an InfoNCE-based objective, and reconstruction loss is measured as channels are incrementally added. As shown in Figure 3b, reconstruction loss decreases monotonically as more channels are incorporated, a trend that holds across different channel orderings. This behavior indicates that each channel contributes complementary information and that no single channel alone suffices to recover the full structural embedding.

**Summary.** Together, these analyses demonstrate that the proposed factorization yields knowledge channels that are specific, weakly dependent, and jointly information-complete. These properties support the interpretability and robustness of ProtDiS and motivate its improved performance in downstream tasks.

### 4.3. Knowledge-Enhanced Functional Prediction

#### 4.3.1. DECOMPOSITION-FUSION IMPROVES ESM-3 PRETRAINED STRUCTURAL ENCODINGS

To evaluate whether our decomposition-fusion strategy enhances pretrained structural embeddings, we compared our method against the ESM-3 Structural Tokenizer (ESM3ST) across twelve downstream tasks under two dataset split schemes: random, and structure-based (Detailed benchmark setups are provided in Section A.3.). For each task, we identified the most relevant knowledge vectors based on the

*Table 1.* Benchmark comparison across different data splits. All results are reported in percentage (%). Best performance within each split is highlighted in bold.

| Split | Model | EC (acc) | MF (fmax) | BP (fmax) | CC (fmax) | Pfam (acc) | SCOP-fa (acc) | SCOP-cf (acc) | SCOP-cl (acc) | Struct. Sim. (spr) | Lig. Aff. (spr) | PPIs (auroc) | Lig. BS. (mcc) |
|---|---|---|---|---|---|---|---|---|---|---|---|---|---|
| Random | Esm3ST | 88.2 | 77.5 | 60.0 | 62.3 | 79.5 | 65.6 | 80.9 | 91.9 | 73.0 | 68.8 | 89.2 | 79.3 |
| | **ProtDiS** | **89.0** | 77.2 | **60.0** | 62.2 | **81.1** | **66.9** | 80.9 | **92.5** | 72.4 | **68.9** | **91.0** | **79.4** |
| Struct. | Esm3ST | 78.7 | 61.1 | 40.3 | 46.2 | 58.9 | 75.0 | 86.1 | 94.6 | 66.7 | 35.1 | 82.1 | 61.7 |
| | **ProtDiS** | **83.5** | **61.2** | **40.5** | **47.1** | **59.5** | **78.0** | **86.6** | **96.0** | **66.9** | **36.6** | **84.6** | **62.3** |

feature-level importance analysis and fused them through a gated network before feeding them into a graph neural network (GNN) for downstream prediction (Detailed procedures are provided in Section B). For instance, in the enzyme class (EC) prediction task, only four knowledge dimensions—residual features, secondary structure, local packing, and contact entropy—were selected and fused. This adaptive combination allows the model to integrate task-relevant biophysical knowledge while suppressing less informative features.

Across the benchmark (Table 1), our approach achieves notable improvements on the majority of datasets, yet exhibits moderate gains or neutral results in a few cases. We attribute this to two main factors. First, datasets with limited sample sizes and large class diversity (e.g., SCOP-cf) are more susceptible to overfitting when incorporating multiple high-dimensional features. Second, certain tasks (e.g., Structure Similarity Prediction) inherently depend on global structural topology rather than localized residue-level knowledge, reducing the benefit of our fine-grained decomposition.

Notably, the performance improvement is most pronounced under structure-based splits, where the test proteins share low structural similarity with the training set. This targeted fusion consistently improved performance over the ESM-3 structural tokenizer when evaluated under identical downstream task architectures: enzyme class prediction increased by 6.05%, ligand-binding affinity prediction by 4.45%, and SCOP family classification by 3.91%. These systematic gains indicate that ProtDiS may encode fine-grained biophysical variations even among proteins with highly similar folds—a hypothesis we explore in the following analyses.

### 4.3.2. KNOWLEDGE-GUIDED DIFFERENTIATION OF STRUCTURALLY SIMILAR ENZYMES

To evaluate whether ProtDiS captures functional distinctions beyond global structural similarity, we conduct a large-scale analysis on enzyme pairs. We randomly sample 100,000 protein pairs from 15,057 enzymes, with balanced positive pairs sharing identical enzyme annotations and negative pairs with distinct functions.

**Function Prediction under High Structural Similarity.** We evaluate functional discrimination by training an XG-Boost classifier to predict whether two proteins share the same function using either structural embeddings or ProtDiS knowledge embeddings. Performance is reported across TM-score bins reflecting increasing structural similarity. As shown in Figure 4a, classifiers based on structural embeddings degrade as TM-score increases, whereas knowledge-guided embeddings remain robust. In the highest TM-score bin, ProtDiS achieves an AUC of 0.946 compared to 0.868 for ESM-3 structural embeddings, indicating improved discrimination among structurally similar but functionally distinct enzymes.

**Mechanistic Analysis via Embedding Similarity.** To understand the source of this improvement, we analyze the relationship between embedding similarity and structural similarity for negative protein pairs with TM-score > 0.5. As shown in Figure 4b, cosine similarity between structural embeddings increases rapidly with TM-score, indicating representation collapse under high structural resemblance. In contrast, knowledge embeddings maintain lower cosine similarity even at high TM-scores, suggesting that ProtDiS preserves fine-grained, function-relevant variation beyond global structural alignment.

**Understanding the Source of Improved Discrimination.** To investigate why knowledge embeddings exhibit stronger discriminative power, we analyze the relationship between embedding similarity and structural similarity. Focusing on negative protein pairs with high structural resemblance (TM-score > 0.5), we examine how cosine similarity varies with TM-score for both structural and knowledge embeddings.

**Qualitative Visualization of Knowledge-specific Differences.** We further provide qualitative visualizations of two representative negative protein pairs to illustrate how ProtDiS highlights interpretable local differences. The first example compares PDB structures 3E9Y and 4JUD (TM-score = 0.823), where secondary-structure annotations reveal subtle but functionally relevant local conformational differences emphasized by the corresponding knowledge channel. The second example compares 2HWT and 2N7Y

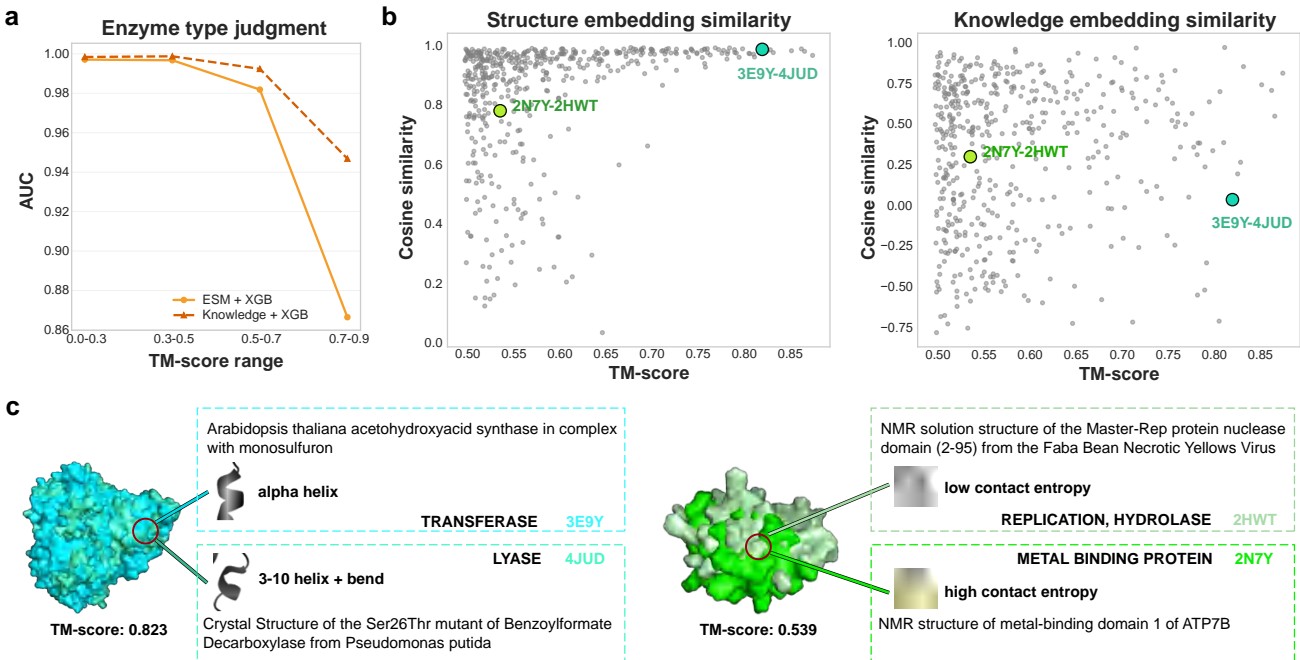

*Figure 4.* Knowledge-aware representations improve discrimination of structurally similar proteins. (a) AUC of EC-label consistency prediction using XGBoost, stratified by TM-score. Knowledge embeddings outperform structural embeddings at high TM-scores. (b) Cosine similarity versus TM-score for protein pairs, showing increased dispersion of knowledge embeddings under high structural similarity. (c) Example protein pairs with similar structures but different functions, where knowledge embeddings capture functionally relevant differences more clearly.

(TM-score = 0.539), where residue-level contact entropy exposes differences in local interaction complexity that remain obscured in structural embeddings despite high cosine similarity. These examples illustrate that ProtDiS captures interpretable micro-environment variations that explain its improved discrimination under strong structural similarity.

## 5. Discussion

In conclusion, ProtDiS provides an interpretable and biophysically grounded framework for structural protein modeling, disentangling micro-environmental signals into meaningful dimensions that preserve predictive power while revealing functional distinctions obscured in conventional embeddings. By disentangling structural encodings into interpretable knowledge dimensions, ProtDiS not only enhances transparency in model behavior but also provides a computational lens for exploring structure–function relationships that were previously obscured in deep embeddings. This suggests a broader role for knowledge-guided representation learning in driving discovery within protein science, where mechanistic insights often remain elusive.

Nevertheless, the current formulation of ProtDiS relies heavily on protein structural data, which limits its applicability

to proteins lacking experimentally determined or accurately predicted structures. This dependence constrains its deployment in large-scale proteome analyses or early-stage discovery pipelines where only sequence-level information is available. Future research will thus focus on extending ProtDiS to sequence and evolutionary feature spaces, enabling the estimation of local structural knowledge directly from amino acid context, co-evolutionary statistics, or learned embeddings from protein language models (Su et al., 2025a; Pantolini et al., 2025).

While this study provides an initial validation of knowledge-guided decomposition, the broader potential of ProtDiS remains largely untapped. Future directions include: (i) integrating ProtDiS into multimodal protein language models, where it can serve as an interpretable structural prior to enhance cross-modal alignment between sequence, structure, and function (Hayes et al., 2025); and (ii) leveraging ProtDiS for knowledge-guided protein design (Lobzaev & Stracquadanio, 2024), where individual knowledge dimensions can be explicitly modulated to control properties such as flexibility, packing density, or hydrophobicity. Collectively, these directions highlight the promise of knowledge-decomposed representations as a foundation for the next generation of interpretable and hypothesis-driven computa-

tional protein science.

## Impact Statement

This paper presents work whose goal is to advance the field of Machine Learning. There are many potential societal consequences of our work, none which we feel must be specifically highlighted here.

## Acknowledgments

This work was supported by Guangdong S&T Programme (Grant No.2024B0101010003). The authors thank Shanghai Smart Logic Technology Co. Ltd. for the valuable support during this research.

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

# A. Implementation Details

## A.1. Model Architecture

The proposed ProtDiS framework is implemented as a **Knowledge Orthogonal Network**, designed to disentangle protein structural embeddings into biologically interpretable and statistically independent subspaces. As shown in Figure 5, the network architecture comprises a *Common Knowledge Network*, multiple *Knowledge-Specific Networks*, and a set of *regularization constraints* that jointly enforce specificity, independence, and completeness.

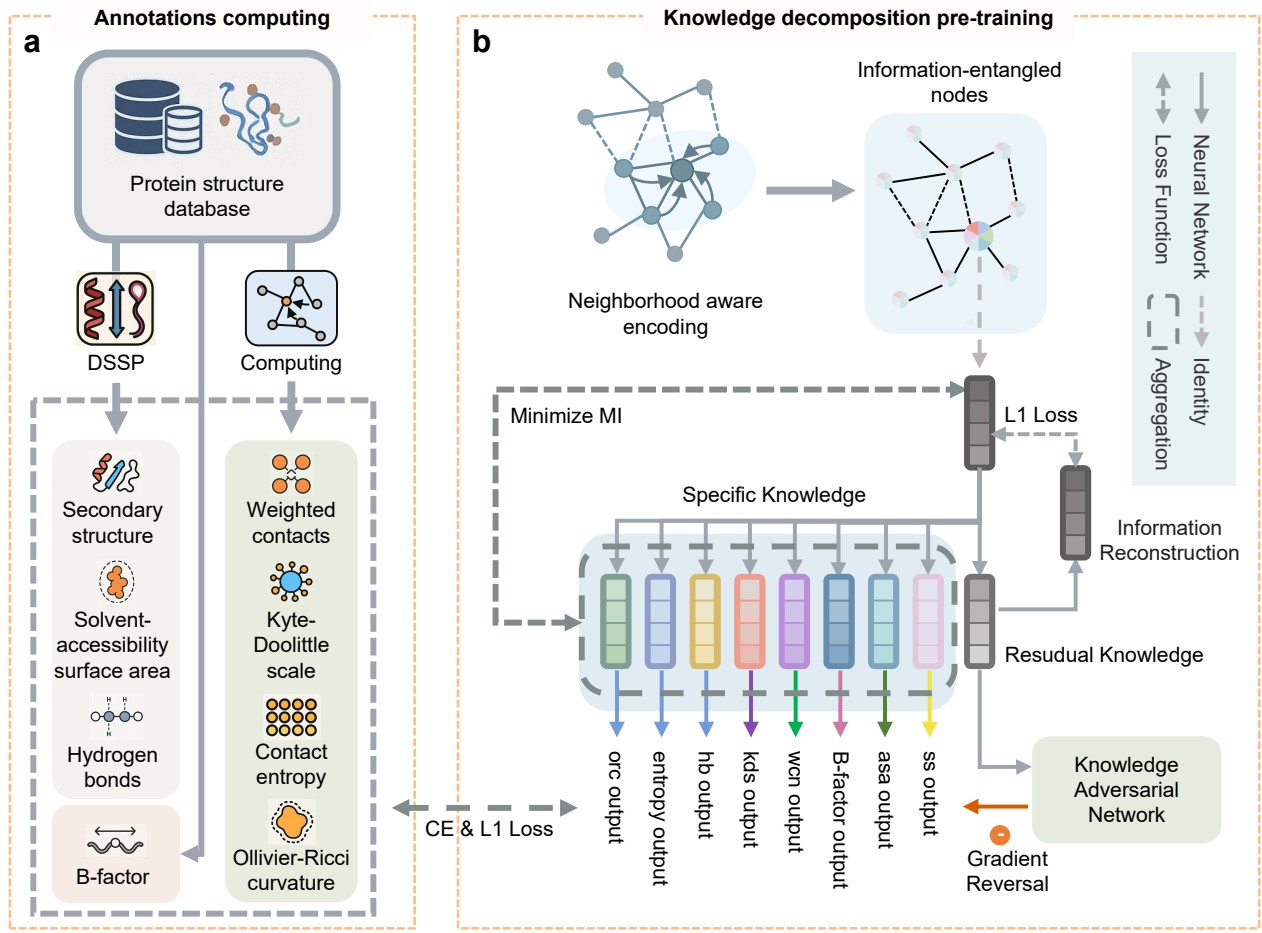

*Figure 5.* **Main architectures of ProtDiS. a**, Pipeline for computing structural descriptors. Core structural features are obtained using DSSP, including secondary-structure assignments, solvent-accessible surface area, and hydrogen-bond energies. Additional microenvironmental properties are computed from atomic coordinates using Python/Biopython, yielding B-factors, weighted contact number, Kyte–Doolittle hydropathy, contact entropy, and Ollivier–Ricci curvature. Together, these descriptors span major biophysical categories—shape, exposure, stability, flexibility, packing density, hydrophobicity, complexity, and curvature—providing a detailed representation of local structural organization. **b**, Model architecture. The network is trained with supervision from the computed structural labels, while multiple regularization objectives are applied to enforce specificity, independence, and completeness of the learned embeddings.

### A.1.1. INPUT AND SHARED ENCODING

Each residue is represented by a 128-dimensional embedding $\mathbf{x}_i \in \mathbb{R}^{128}$ extracted from the pretrained *ESM-3 Structural Tokenizer*, encoding the local geometric and physicochemical context.

These embeddings are processed by a two-stage **Common Knowledge Network**:

$$\mathbf{c}_1 = f_{\text{ckn}_1}(\mathbf{x}), \qquad \mathbf{c}_2 = f_{\text{ckn}_2}(\mathbf{c}_1), \tag{21}$$

where each sub-network $f_{\text{ckn}_i}(\cdot)$ is a multi-layer perceptron (MLP) followed by layer normalization:

$$f_{\text{ckn}_i}(\mathbf{h}) = \text{LN}\left(W_i^{(2)}\sigma(W_i^{(1)}\mathbf{h} + b_i^{(1)}) + b_i^{(2)}\right), \tag{22}$$

and $\sigma(\cdot)$ denotes the SwiGLU activation. The outputs $\mathbf{c}_1$ and $\mathbf{c}_2$ form the hierarchical *common knowledge representations*. A detached copy $\mathbf{c}_2^{adv} = f_{\text{ckn}_2}(\text{stopgrad}(\mathbf{c}_1))$ is used for adversarial disentanglement.

### A.1.2. KNOWLEDGE-SPECIFIC NETWORKS

The model constructs eight parallel **Knowledge-Specific Networks**:

$$\mathcal{F} = \{\text{SS}, \text{ASA}, \text{BF}, \text{WCN}, \text{KDS}, \text{HB}, \text{CE}, \text{ORC}\},$$

each corresponding to a biologically grounded descriptor—secondary structure (SS), solvent-accessible surface area (ASA), local flexibility (BF), packing density (WCN), hydrophobicity (KDS), hydrogen-bond strength (HB), side-chain conformational entropy (CE), and Ollivier–Ricci curvature (ORC).

For each $f_i \in \mathcal{F}$, a branch projects the shared latent $\mathbf{c}_1$ into a feature-specific subspace:

$$\mathbf{s}_i = f_{\text{ksn}_i}(\mathbf{c}_1) = W_i^{(2)}\sigma(W_i^{(1)}\mathbf{c}_1 + b_i^{(1)}) + b_i^{(2)}. \tag{23}$$

Each knowledge branch includes a prediction head $g_i(\cdot)$, implemented as a two-layer MLP with ReLU activation and layer normalization:

$$\hat{y}_i = g_i(\mathbf{s}_i) \tag{24}$$

The supervision type depends on the knowledge label:

$$\mathcal{L}_{fea_i}^{(cls)} = -\frac{1}{N}\sum_{n=1}^{N} y_{i,n}\log\hat{y}_{i,n}, \qquad\qquad \text{for discrete labels (e.g., SS);} \tag{25}$$

$$\mathcal{L}_{fea_i}^{(reg)} = \frac{1}{N}\sum_{n=1}^{N} |y_{i,n} - \hat{y}_{i,n}|, \qquad\qquad \text{for continuous labels (e.g., HB, KDS).} \tag{26}$$

### A.1.3. OUTPUT REPRESENTATIONS

The final outputs include:

- $\mathbf{c}_2$: the **common knowledge representation**;
- $\{\mathbf{s}_i\}_{i=1}^{8}$: the **knowledge-specific embeddings**;

These embeddings can be selectively fused for downstream tasks such as enzyme classification, binding site prediction, and knowledge-guided protein design.

### A.1.4. TRAINING PROCEDURE

The model was trained for 300 epochs using the AdamW optimizer with $\beta_1 = 0.9$, $\beta_2 = 0.95$, and a weight decay of $1 \times 10^{-2}$. The initial learning rate was set to $1 \times 10^{-3}$ and dynamically adjusted by a two-stage schedule combining linear warmup (first 1,000 steps) and cosine decay, which gradually reduced the rate to 0.01 of its peak value. The adversarial branches, when enabled, used a scaled learning rate twice as high as the main encoder to stabilize disentanglement dynamics. Gradient clipping (maximum norm 5.0) was employed to suppress potential exploding gradients during the early training stage.

Each epoch consisted of a full forward–backward pass through the dataset with a batch size of 64. The total loss combined supervised knowledge prediction, reconstruction, and regularization terms. The best-performing model was selected based on the monitoring of supervised loss components.

## A.2. Data Preprocessing

### A.2.1. TRAINING DATA COLLECTION

To construct the dataset for knowledge-guided structural representation learning, we integrated experimentally determined and computationally predicted protein structures from multiple public sources. Specifically, we collected all available protein structures from the **Protein Data Bank (PDB, version 20250124)** (Burley et al., 2024) and from the **AlphaFold Protein Structure Database (AlphaFold–SWISS-PROT release)** (Fleming et al., 2025).

For the PDB dataset, we retained only entries with an average atomic B-factor less than 80 Å$^2$ to ensure structural quality. For the AlphaFold–SWISS-PROT dataset, we filtered structures with a predicted local confidence (pLDDT) greater than 90. After quality filtering, we randomly sampled 50,000 protein complexes from the PDB dataset (keeping only their protein components) and 50,000 individual polypeptide chains from the SWISS-PROT dataset. This downsampling was motivated by computational efficiency and statistical sufficiency: since the proposed knowledge-decomposition framework operates at the *micro-environment level*, 100,000 distinct protein chains already provide forty of millions of local residue environments, which is sufficient for model convergence and eneralization.

All structures were standardized using BioPython and PyMOL-based preprocessing, including removal of heteroatoms and alternate positions, assignment of missing residues via ModRefiner reconstruction, and coordinate normalization to a global Cartesian reference frame. Hydrogen atoms were ignored for consistency across experimental and predicted data.

### A.2.2. KNOWLEDGE LABEL COMPUTATION

For each residue in every protein file, we computed eight biologically interpretable structural knowledge labels representing distinct micro-environmental dimensions. These labels serve as explicit supervisory signals guiding representation disentanglement.

**Shape.** The backbone conformation of each residue $r_i$ was assigned using the DSSP algorithm (Kabsch & Sander, 1983b; Hekkelman et al., 2025):

$$ss_i = \text{DSSP}(r_i),$$

yielding one of eight canonical secondary structure classes (H, E, G, I, T, S, B, C).

**Exposure.** DSSP-derived (Hekkelman et al., 2025) solvent-accessible surface area was recorded as:

$$ASA_i = A_i^{\text{abs}}, \qquad RASA_i = \frac{A_i^{\text{abs}}}{A_i^{\text{max}}},$$

where $A_i^{\text{max}}$ denotes the maximal accessibility of the amino acid type (Tien et al., 2013).

**Flexibility.** Residue-level flexibility was quantified as the mean atomic B-factor (Savojardo et al., 2017):

$$B_i = \frac{1}{|\mathcal{A}_i|} \sum_{a \in \mathcal{A}_i} B_a,$$

with $\mathcal{A}_i$ being the set of heavy atoms within residue $i$.

**Packing density.** We defined local density density using the Weighted Contact Number (WCN) (Marcos & Echave, 2015):

$$\text{WCN}_i = \sum_{\substack{j \neq i \\ r_{ij} < 20\text{Å}}} \frac{1}{r_{ij}^2},$$

where $r_{ij}$ is the C$\alpha$–C$\alpha$ distance between residues $i$ and $j$.

**Hydrophobicity.** The local hydrophobic environment was estimated as the average Kyte–Doolittle hydrophobicity (Kyte & Doolittle, 1982) of contacting residues within a 4.5 Å cutoff:

$$H_i = \frac{1}{|\mathcal{N}_i|} \sum_{j \in \mathcal{N}_i} h(a_j),$$

where $h(a_j)$ is the scale value for amino acid $a_j$.

**Stability.** Hydrogen-bond-derived stability was calculated from hydrogen-bond statistics (Kabsch & Sander, 1983a):

$$S_i = 0.4\rho_{\text{hb}} + 0.5\bar{E}_{\text{hb}} + 0.1r_{\text{strong}},$$

where $\rho_{\text{hb}}$ is the hydrogen bond density, $\bar{E}_{\text{hb}}$ is the mean absolute hydrogen-bond energy, and $r_{\text{strong}}$ is the fraction of strong bonds with $E < -1.5\,\text{kcal/mol}$. The original information was obtained using DSSP (Hekkelman et al., 2025).

**Complexity.** Local topological diversity was expressed by the normalized Shannon entropy of the contact-degree distribution (Huang et al., 2016):

$$H_i = -\frac{1}{\log|\mathcal{N}_i|} \sum_k p_k \log p_k, \quad p_k = \frac{n_k}{\sum_k n_k},$$

where $n_k$ is the number of residues in the contact subgraph with degree $k$.

**Curvature.** Finally, we measured the local geometric curvature with Ollivier–Ricci Curvature (ORC) (Zheng et al., 2025) as:

$$\kappa(x,y) = 1 - \frac{W_1(p_x, p_y)}{d(x,y)}, \quad \text{ORC}_i = \frac{1}{|\mathcal{N}_i|} \sum_{y \in \mathcal{N}_i} \kappa(i,y),$$

where $W_1(p_x, p_y)$ denotes the Wasserstein-1 distance between random-walk distributions on neighboring residues $x$ and $y$, and $d(x,y)$ is the Euclidean C$\alpha$–C$\alpha$ distance.

The selection of these specific eight dimensions is governed by a balance of biological relevance, mutual independence, and computational feasibility:

**1. Scale Consistency and Functional Relevance:** The chosen dimensions are strictly constrained to a unified, residue-level scale, encompassing local geometric and physicochemical properties. This design deliberately avoids scale-mismatch issues that would arise from integrating macroscopic (e.g., global fold classes) or microscopic (e.g., quantum mechanical) labels. Biologically, these local properties serve as the first principles that dictate higher-order 3D conformations and downstream functional behaviors.

**2. Relative Independence:** To ensure that the selected dimensions provide distinct supervisory signals, we evaluated their relative independence by computing the Mutual Information (MI) among all eight labels. As illustrated in Figure 6, the off-diagonal MI values remain predominantly low (for instance, the MI between Secondary Structure and other labels is $0.012 \sim 0.210$). This low redundancy is critical, as it forces the network to map representations into divergent and strictly disentangled latent subspaces.

**3. Computational Feasibility:** Given the massive scale of the requisite pretraining datasets (encompassing structures from both the PDB and AlphaFoldDB), the empirical viability of our framework heavily relies on highly scalable supervision. The selected eight dimensions can be deterministically and rapidly computed using standardized algorithms (e.g., DSSP) with high confidence. This explicitly avoids the prohibitive computational overhead associated with complex profiling methods, such as Molecular Dynamics (MD) simulations, thereby enabling efficient large-scale training.

### A.3. Benchmark and Evaluation

#### A.3.1. TASK-BY-TASK DATA SOURCES AND COMPOSITION

The downstream benchmarks adopted in this study are derived from the **ProteinShake** framework (Kucera et al., 2023), which integrates diverse public databases to construct standardized protein structure learning tasks. We summarize below the eight task families and their corresponding data sources used in our evaluation. Each task reflects a distinct structural or functional aspect of proteins.

**Enzyme Class (EC)** The enzyme classification task is based on the **Enzyme Commission (EC)** hierarchy curated in UniProt/Swiss-Prot. We exposes the top three EC levels as prediction targets, covering seven major enzymatic classes (*Oxidoreductases*, *Transferases*, *Hydrolases*, *Lyases*, *Isomerases*, *Ligases*, and *Translocases*). The dataset contains approximately 15,600 annotated proteins.

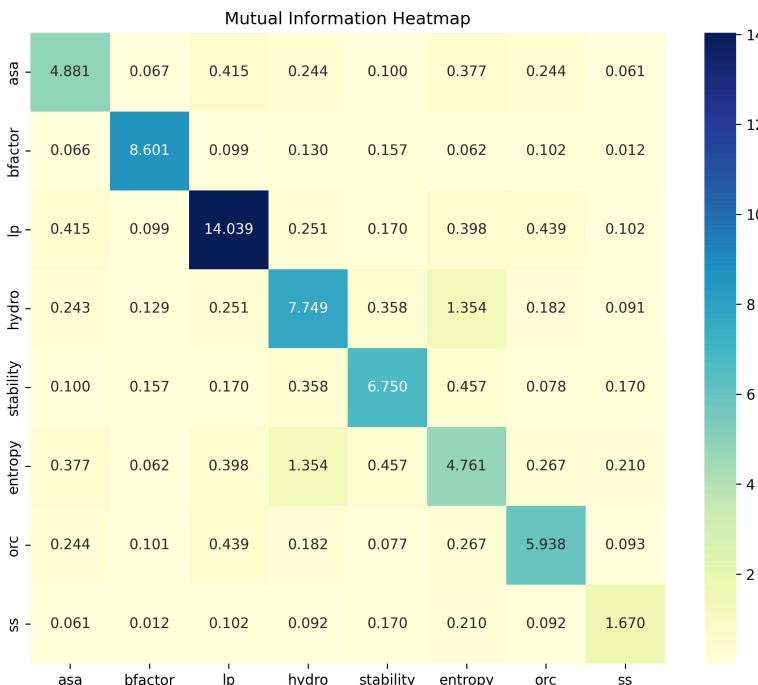

*Figure 6.* Mutual Information (MI) heatmap among the eight selected structural and physicochemical labels, demonstrating low off-diagonal redundancy.

**Gene Ontology (GO)**   The Gene Ontology dataset compiles functional annotations from UniProt/Swiss-Prot and maps them to the **GO hierarchy**, which describes proteins from three complementary biological perspectives: *molecular function (MF)*, *biological process (BP)*, and *cellular component (CC)*. Following ProteinShake, we treat GO prediction as a multi-label classification problem, and in our experiments, we evaluate these three GO aspects as independent subtasks. The dataset comprises about 32,600 proteins with curated GO annotations.

**Protein Family (Pfam)**   This task leverages the **Pfam** database, which groups proteins into evolutionarily related families based on conserved domains and sequence motifs. Each protein chain is assigned a Pfam family label, and the objective is to classify a protein into its corresponding family. The dataset contains over 31,000 proteins and is formulated as a multi-class classification problem.

**Structure Class (SCOP)**   The structural classification task originates from the **Structural Classification of Proteins (SCOP)** database, which organizes proteins into a hierarchical taxonomy according to their structural folds. To probe the model's sensitivity to different structural resolutions, we decompose this task into three subtasks of increasing granularity:

- **scop-cl:** classification at the *Class* level, distinguishing broad categories of secondary-structure organization;

- **scop-cf:** classification at the *Fold* level, focusing on topological and spatial arrangement of structural motifs;

- **scop-fa:** classification at the *Family* level, capturing fine-grained distinctions within evolutionarily related groups.

Together, these subtasks assess the model's capacity to recognize global secondary-structure types, identify shared topological folds, and resolve subtle structural variations within homologous families.

**Structural Similarity**   This regression task evaluates the ability of a model to predict structural similarity between two protein chains. ProteinShake constructs protein pairs by sampling and aligning single-chain structures using TM-align, with the local distance difference test (LDDT) score serving as the ground-truth similarity measure. Given two unaligned structures, the model must predict their aligned LDDT score, and performance is measured by Spearman rank correlation.

**Ligand Affinity**    The ligand affinity prediction task uses the **PDBBind-CN** database, which contains experimentally determined protein–ligand complexes and corresponding binding affinities. Given a protein structure and its ligand (represented as a SMILES string), the goal is to predict the quantitative binding affinity (e.g., $K_d$ or $pK_d$). Performance is evaluated by the Spearman rank correlation between predicted and measured affinities.

**Protein–Protein Interface**    This task focuses on inter-chain contact prediction using protein complex data aggregated from the PDB. For each complex, ProteinShake provides ground-truth residue–residue contact matrices derived from interface geometry (contacts are defined by a $C\alpha$-$C\alpha$ distance cutoff of 6.0 Å). The model is trained to classify residue pairs as contacting or non-contacting, and performance is evaluated using the area under the receiver operating characteristic curve (AUROC).

**Binding Site Detection**    The binding site prediction task is based on the pocket annotations from **PDBBind**, which identify residues located within experimentally observed ligand-binding pockets. For each residue or residue-centered environment, the model predicts whether it belongs to a binding site. Performance is assessed using Matthew's correlation coefficient (MCC), which balances sensitivity and specificity in imbalanced datasets.

**Summary: total number of sub-tasks**    In total, the eight task families from ProteinShake yield twelve supervised downstream tasks in our experiments: **EC**; **GO-MF**, **GO-BP**, **GO-CC**; **Pfam**; **scop-cl**, **scop-cf**, **scop-fa**; **Structural Similarity**; **Ligand Affinity**; **Protein–Protein Interface**; and **Binding Site Detection**. Together, these benchmarks comprehensively cover protein structure, function, and interaction prediction, forming a rigorous evaluation suite for structure-informed representation learning.

### A.3.2. BENCHMARK CONSTRUCTION AND SPLIT STRATEGIES

ProteinShake standardizes dataset construction through three steps: (i) raw data retrieval from primary sources, (ii) quality filtering and annotation aggregation, and (iii) dataset conversion to representations required by models (graphs, point clouds, voxels). Splits are pre-computed and versioned, enabling reproducibility. Key details:

- **Random split:** standard IID split into training/validation/test sets.

- **Sequence-similarity split:** cluster proteins by sequence similarity (CD-HIT or similar) and allocate whole clusters to splits so that similar sequences fall into the same partition (controls sequence-homology leakage). Thresholds used in ProteinShake can be varied (30%–90%); default experiments often use 70% similarity cutoff.

- **Structure-similarity split:** cluster by structural similarity using Foldseek (or structural alignment) and LDDT thresholds; clusters are formed iteratively by sampling a protein and retrieving structurally similar instances. This split is the most stringent (tests true structural OOD generalization). ProteinShake uses default structure-similarity thresholds in the 50%–90% range (70% commonly used in reported experiments).

ProteinShake provides precomputed splits for each task under random, sequence, and structure splitting regimes; we adopt the same split types for fair comparisons. Performance typically degrades from random → sequence → structure splits, indicating increased OOD difficulty.

### A.3.3. EVALUATION METRICS: DEFINITIONS AND CALCULATION

ProteinShake designates a default metric per task; below we present the computation of each relevant metric used in our experiments. All metrics are standard; we provide formulas for clarity.

**Accuracy (multi-class)**    For a dataset with $N$ samples and true class labels $y_n$ and predicted class $\hat{y}_n$,

$$\text{Accuracy} \;=\; \frac{1}{N} \sum_{n=1}^{N} \mathbb{I}(\hat{y}_n = y_n),$$

where $\mathbb{I}$ is the indicator function. Used for Enzyme Class, Pfam, Structure Class variants.

**F$_{\max}$ (for multi-label GO)**  The F$_{\max}$ score is the maximal F1 score over a sweep of prediction thresholds (commonly used for multi-label function prediction). For a threshold $t$, per-protein precision and recall are computed and averaged (or computed globally depending on implementation). Denote $P(t)$ and $R(t)$; then

$$\mathrm{F1}(t) = \frac{2P(t)R(t)}{P(t) + R(t)}, \qquad \mathrm{F_{max}} = \max_{t} \mathrm{F1}(t).$$

ProteinShake uses F$_{\max}$ as the default for GO tasks.

**AUROC (Area Under the ROC curve)**  For binary/residue-level or matrix predictions (e.g., interface contact predictions), AUROC is computed by ranking predicted scores and computing area under the true positive rate vs false positive rate curve. When multiple matrices/pairs exist, ProteinShake reports the median AUROC across instances as the default summary.

**Matthew's Correlation Coefficient (MCC)**  For binary classification (binding-site detection), MCC is computed from confusion matrix entries $TP, TN, FP, FN$ as

$$\mathrm{MCC} = \frac{TP \cdot TN - FP \cdot FN}{\sqrt{(TP + FP)(TP + FN)(TN + FP)(TN + FN)}}.$$

MCC balances class imbalance and is used for binding-site detection in ProteinShake.

**Spearman rank correlation.**  For structural similarity regression and ligand affinity the rank correlation between predicted and true scores is used:

$$\rho = 1 - \frac{6 \sum_{n=1}^{N} (R(\hat{y}_n) - R(y_n))^2}{N(N^2 - 1)},$$

where $R(\cdot)$ denotes rank. Spearman is robust to nonlinear monotone relationships.

### A.3.4. Downstream Predictor Architectures

To evaluate the efficacy of the extracted representations, we design distinct downstream predictor architectures tailored to the specific granularity of the biological tasks, namely protein-level and residue-level predictions. The detailed architectures and the formulation of the final representations are described below.

**Protein-Level Tasks**  For tasks requiring graph-level predictions, such as Enzyme Commission (EC) number prediction, Gene Ontology (GO) term annotation, and Structural Similarity assessment, we employ the Continuous-Discrete Convolution (CDConv) network (Fan et al., 2023) as our primary encoder. The CDConv architecture effectively processes the protein structural graph by aggregating both geometric and sequential contexts. To construct the final protein-level representation, we apply a global pooling operation (utilizing either Average or Max pooling) over the node embeddings extracted from the final CDConv block. This aggregated graph-level embedding is subsequently processed by a Multi-Layer Perceptron (MLP) to yield the final prediction.

**Residue-Level Tasks**  For node-level objectives, such as Ligand Binding Site prediction, we utilize a multi-layer Graph Isomorphism Network (GIN) architecture. Within this framework, each GINConv layer iteratively updates the node embeddings by combining local neighborhood aggregation with an MLP block, specifically structured as Linear $\rightarrow$ BatchNorm1d $\rightarrow$ ReLU $\rightarrow$ Linear. Unlike protein-level tasks, the final representation for these tasks relies directly on the node-level embeddings produced by the terminal GNN layer. These dense, localized representations are directly passed through a task-specific projection head to perform the final residue-level classification.

## B. Knowledge-dependency of Protein Function at the Feature Level

### B.1. Task-Feature Importance Assessment

Leveraging these knowledge-grounded representations, we next examined the dependencies between biophysical dimensions and protein function across twelve downstream tasks (Figure 7), following benchmarking protocols from ProteinShake (Kucera et al., 2023). Lightweight probes trained on individual knowledge channels revealed distinct cross-task patterns:

solvent exposure, packing density, and curvature consistently dominated protein–protein interaction prediction, reflecting geometric and accessibility constraints at molecular interfaces. In contrast, flexibility emerged as a key determinant for enzyme classification and Gene Ontology annotation, consistent with the role of local dynamism in catalytic and regulatory processes (Wang et al., 2023). Notably, the residual "common knowledge" vector—which captures structural information not explicitly assigned to the predefined dimensions—shows substantial importance in several tasks, suggesting contributions from electrostatics, long-range coupling, or other implicit factors.

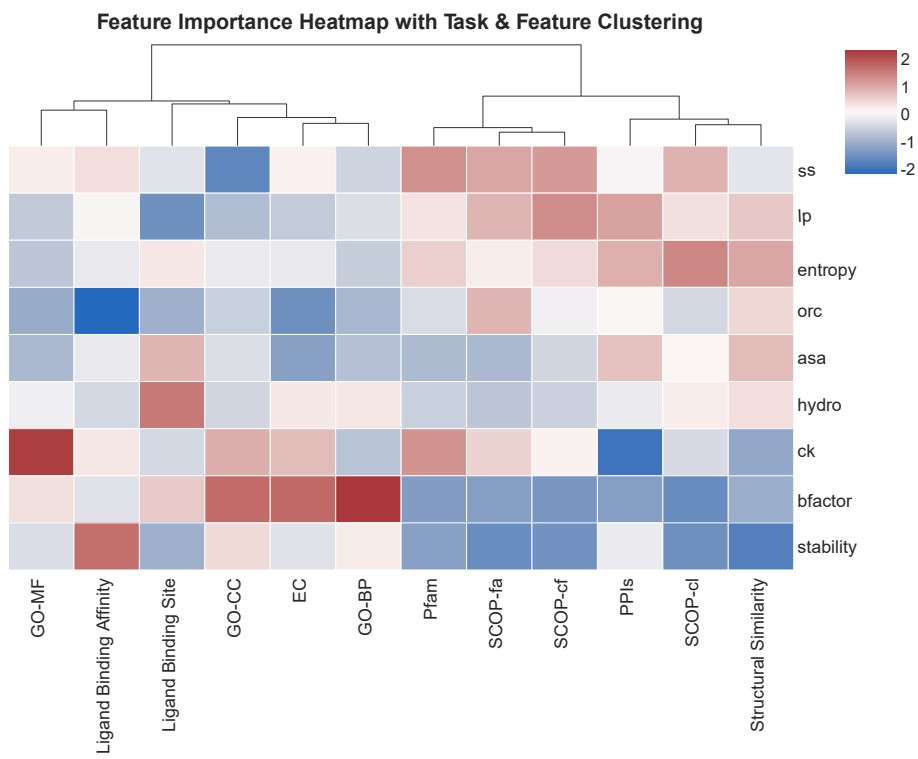

*Figure 7.* Probe-based importance analysis of the eight knowledge dimensions across twelve downstream tasks, revealing distinct biological dependencies.

Beyond single-task analysis, cross-task consistency patterns (Figure 8) further revealed that SCOP structural classes and Pfam families share similar knowledge dependencies, as do enzyme classification and Gene Ontology annotation, indicating that ProtDiS captures coherent functional relationships and may offer a principled route for exploring the shared biophysical determinants underlying diverse protein functions.

### B.2. Feature Gated Fusion

To further assess how the combination of knowledge dimensions affects performance, we conducted a feature accumulation analysis . For selected representative tasks, we incrementally increased the number of input knowledge dimensions in order of their importance and tracked task performance. We found that as the number of features increased, model accuracy initially improved but eventually reached a performance plateau. This saturation effect suggests that low-importance knowledge dimensions contribute little additional useful information and may even introduce noise that hinders generalization.

Motivated by this behavior, we designed a decomposition–fusion strategy (Figure 9) in which only task-relevant channels, determined through importance profiling, are selectively fused for downstream prediction. Typically, we select a subset of 4 to 5 decoupled embeddings to construct the optimal input representation for a given task, thereby filtering out task-irrelevant noise. The exact combinations of embeddings utilized across our 12 evaluated downstream tasks are cataloged in Table 2.

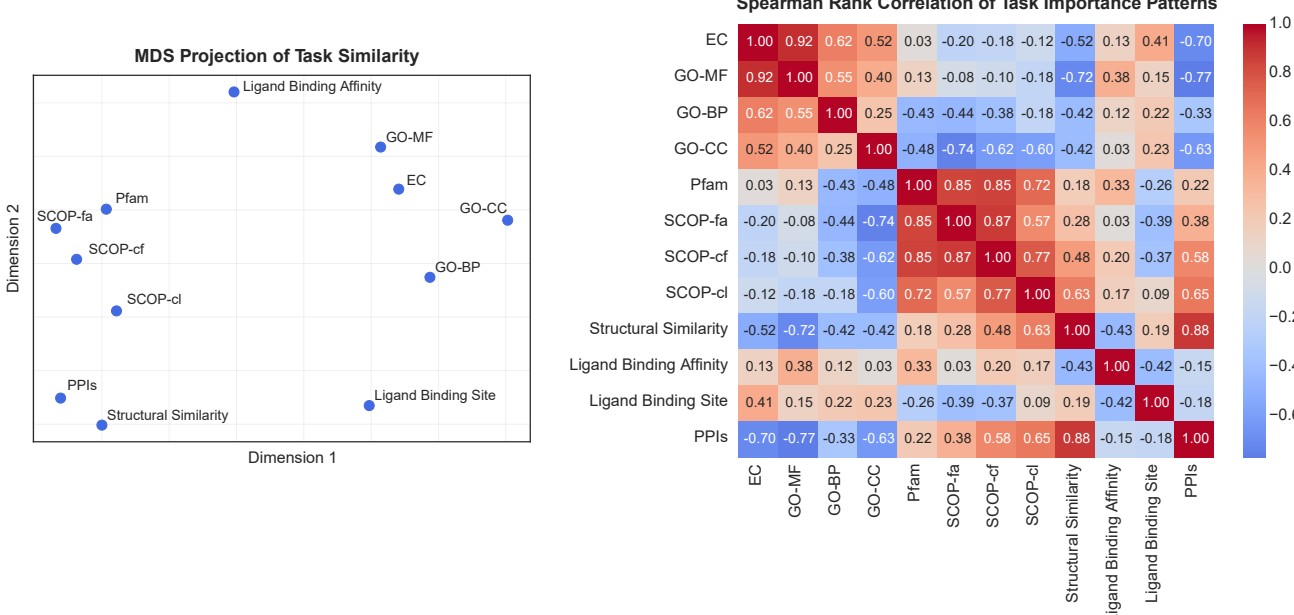

*Figure 8.* Cross-task consistency patterns highlight shared and divergent biophysical drivers across functional groups.

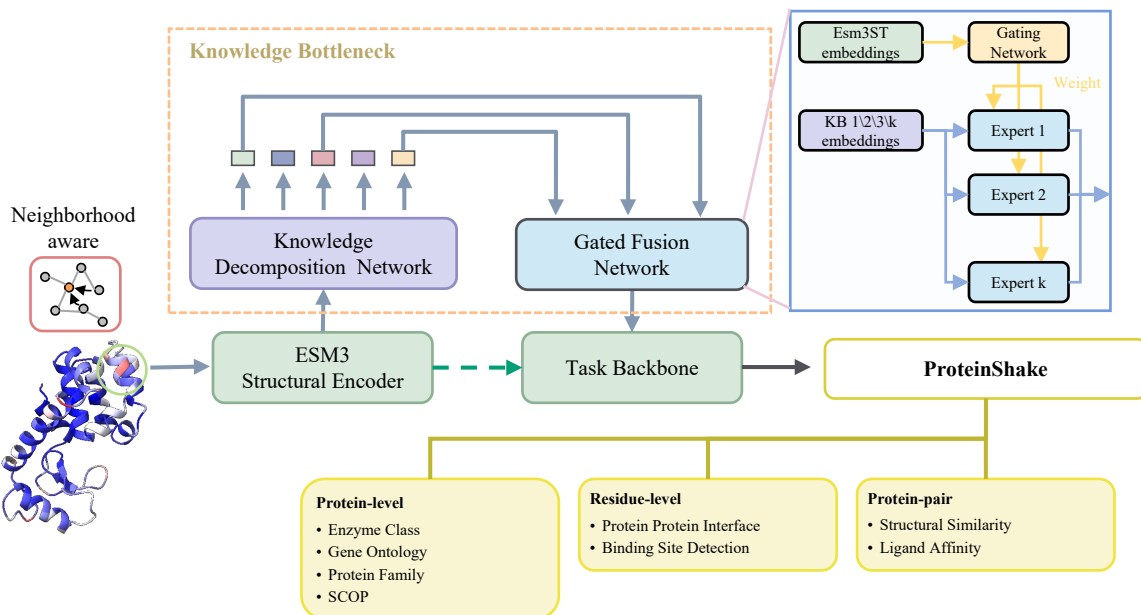

*Figure 9.* Integration of disentangled knowledge features in downstream tasks. Task-relevant knowledge components are selected and fused before being fed into downstream predictors, enabling explicit incorporation of interpretable structural knowledge into diverse functional tasks.

*Table 2.* Task-specific combinations of decoupled structural embeddings utilized in `ProtDiS` across different evaluation levels.

| Task | Level | Selected Embeddings |
|---|---|---|
| EC | Protein-level | residual, shape, flexibility, hydrophobicity |
| GO-MF | Protein-level | residual, shape, flexibility, hydrophobicity, stability |
| GO-BP | Protein-level | flexibility, hydrophobicity, stability |
| GO-CC | Protein-level | residual, flexibility, stability |
| Pfam | Protein-level | residual, shape, packing density, complexity |
| SCOP-fa | Protein-level | residual, shape, packing density, complexity, curvature |
| SCOP-cf | Protein-level | residual, shape, packing density, complexity |
| SCOP-cl | Protein-level | shape, exposure, packing density, hydrophobicity, complexity |
| Structural Similarity | Protein-level | exposure, packing density, hydrophobicity, complexity, curvature |
| Ligand Binding Affinity | Protein-level | residual, shape, packing density, stability |
| Ligand Binding Site | Residue-level | exposure, flexibility, hydrophobicity, complexity |
| PPIs | Residue-level | exposure, packing density, complexity |

## B.3. Enzyme Functional Classification

We extended our analysis to enzyme functional classification involving seven major categories—oxidoreductases, transferases, hydrolases, lyases, isomerases, ligases, and translocases—and computed SHAP values to interpret feature contributions. As illustrated in Figure 10a, the secondary structure, local packing, and contact entropy dimensions consistently exhibited high importance, suggesting that local geometric and conformational properties critically underlie enzymatic differentiation. We further visualized the distributions of these knowledge features across enzyme classes, revealing distinctive patterns; for instance, translocases are enriched in $\alpha$-helical structures and exhibit higher contact entropy, indicating their flexible and loosely packed architectures (Figure 10b,c).

## C. Other Experiments

### C.1. Knowledge-enhanced Characterization of Ligand-binding Sites

To further evaluate the capacity of ProtDiS in resolving fine-grained functional distinctions, we conducted a large-scale residue-level analysis focused on ligand binding sites. Specifically, we sampled approximately 800,000 residue pairs from over 4,000 proteins, where positive pairs corresponded to residues sharing the same functional role (both binding or both non-binding), and negative pairs represented functionally distinct residues. To ensure balance, positive and negative samples were equally distributed.

We first examined the relationship between pairwise similarity and functional consistency. As shown in Figure 11(a, left), the proportion of positive pairs remains roughly constant (around 50%) as structural similarity increases, indicating that structural resemblance alone provides limited information about residue function. In contrast, as knowledge similarity increases, the proportion of positive pairs rises sharply and asymptotically approaches 1, demonstrating that residues sharing similar knowledge embeddings are far more likely to share functional identity. These results suggest that while structural descriptors capture geometric alignment, ProtDiS's knowledge representations encode more functionally meaningful distinctions.

To quantify this observation, we trained XGBoost classifiers to predict whether two residues share the same function, using either structural or knowledge embeddings as inputs. As illustrated in Figure 11(a, right), models based on structural embeddings exhibit a notable performance drop for residue pairs with high structural similarity, whereas those using knowledge embeddings maintain stable accuracy. This contrast implies that knowledge features better preserve discriminability for structurally similar residues, allowing the model to distinguish subtle functional differences among nearly identical binding environments.

Next, we selected two representative examples of structurally similar but functionally divergent sites, and analyzed them using SHAP-based feature attribution. The resulting contribution profiles (Figure 11(b), bar plots) reveal that local packing consistently emerges as the most influential feature, followed by solvent accessibility and secondary structure. To visualize

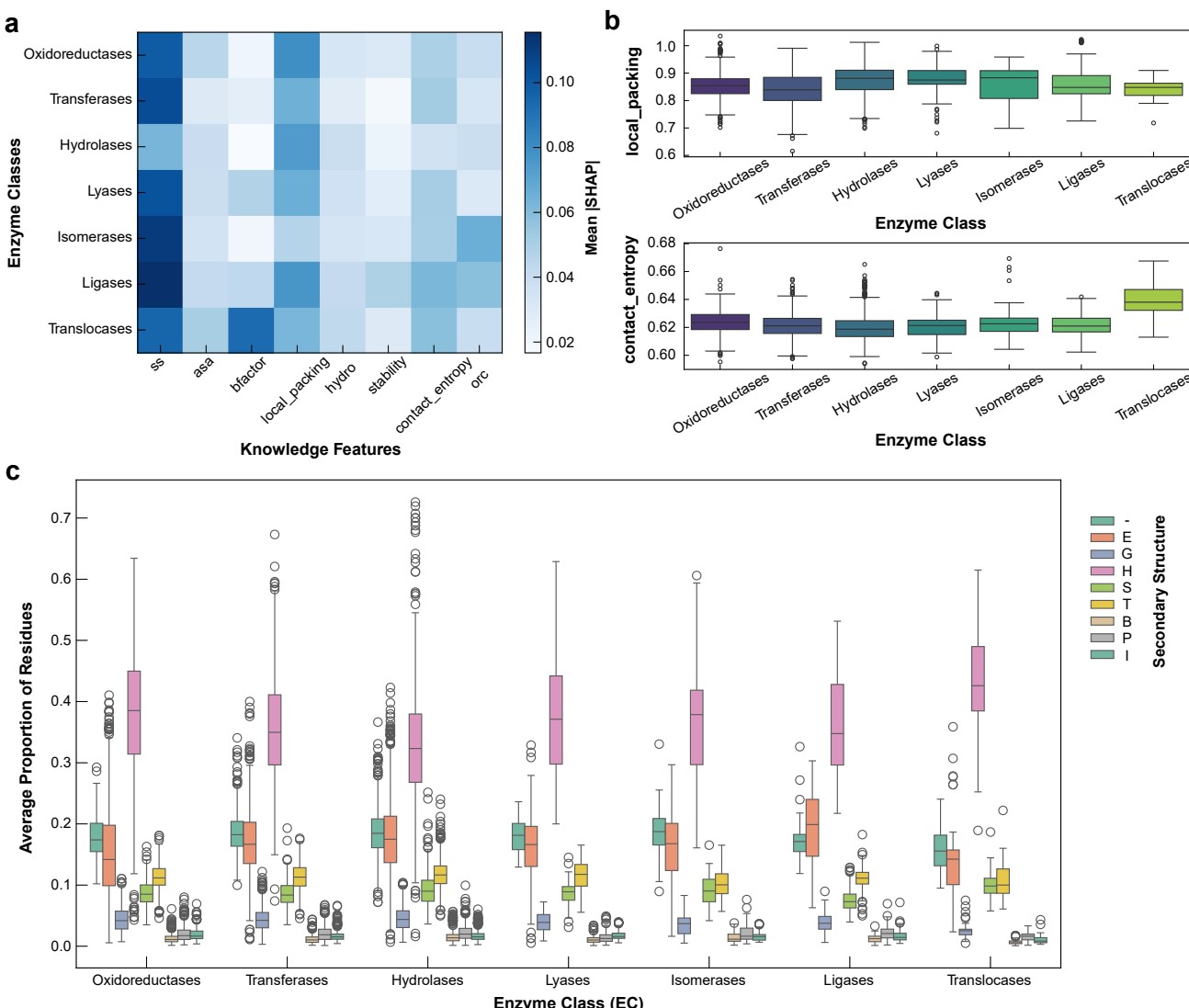

*Figure 10.* SHAP analysis for XGBoost classifiers across seven major UniProt enzyme classes identifies which knowledge dimensions drive class-level distinctions, revealing characteristic biophysical fingerprints.

this insight, we compared the corresponding knowledge radar plots for the two site pairs (Figure 11(b), radar charts), and further projected the local packing values onto their 3D structures. Despite near-identical structural conformations, the knowledge maps display pronounced differences, particularly in packing density and solvent exposure around the binding residues.

These findings collectively demonstrate that ProtDiS disentangles latent knowledge signals critical for site-specific functionality, providing representations that retain high discriminative power even under extreme structural similarity. This property makes ProtDiS a promising foundation for fine-grained tasks such as binding site prediction and functional residue annotation, where structural cues alone often fail to capture subtle yet essential biochemical distinctions.

### C.2. Representation of Highly Variable Regions: Antibody CDRs

A critical question in structural representation learning is how the framework handles highly variable, non-standard regions, such as the Complementarity-Determining Regions (CDRs) in antibodies or nanobodies (VHHs). `ProtDiS` encodes these

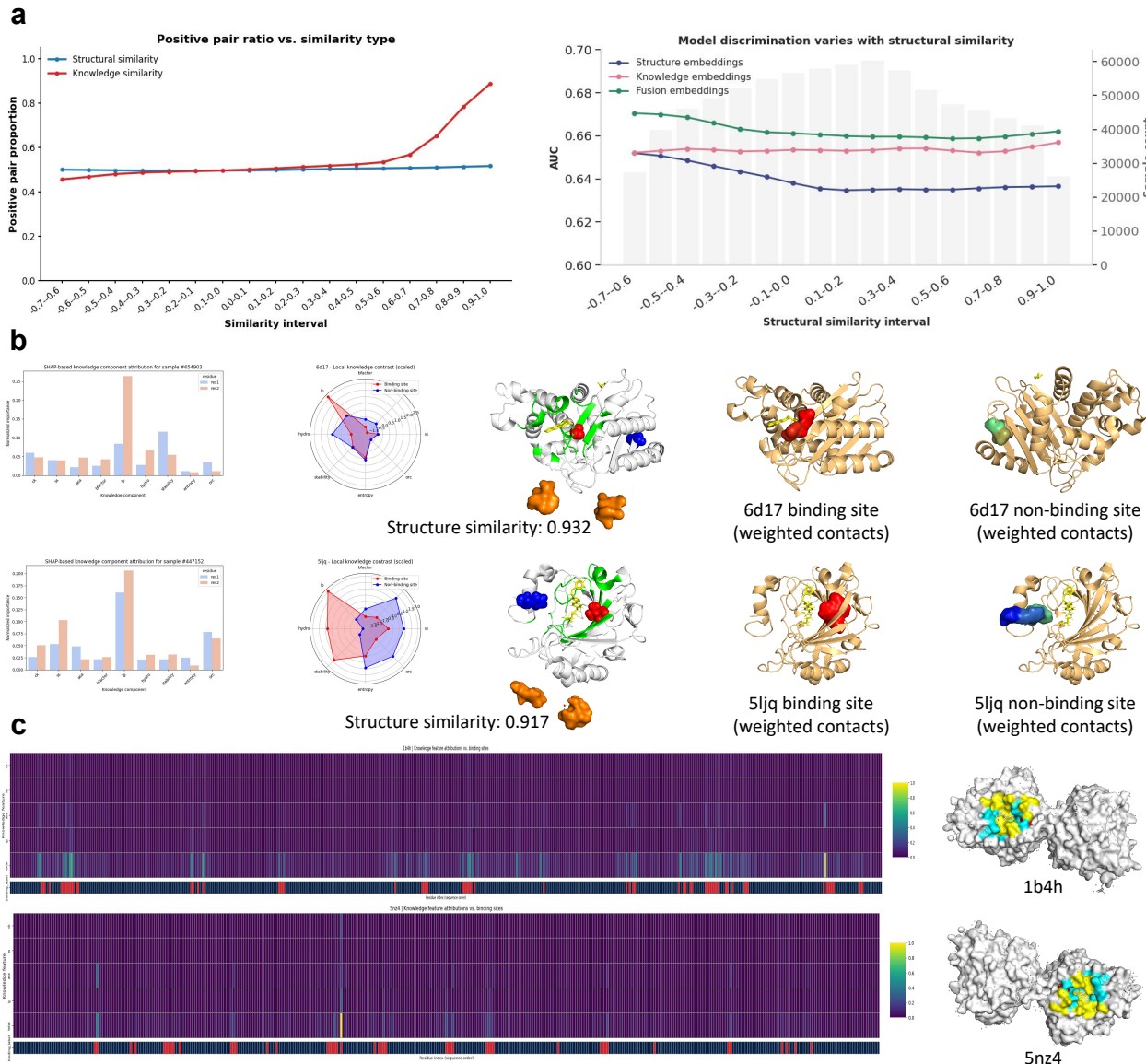

*Figure 11.* **Functional discrimination and interpretability of ProtDiS in protein–ligand binding-site analysis. a**, Functional consistency analysis across 800,000 sampled residue pairs. For knowledge embeddings, similarity monotonically tracks the probability that two residues share the same functional role, approaching unity at high similarity; in contrast, structure-embedding similarity remains functionally agnostic, maintaining a ∼0.5 positive-pair ratio. Consistently, XGBoost classifiers trained to predict functional agreement show marked performance degradation for structure embeddings at high structural similarity, whereas knowledge embeddings retain stable predictive power. **b**, SHAP-based attribution of residue-pair predictions for two representative examples (*PDB 6d17* and *5ljq*). Radar plots illustrate differences in microenvironmental knowledge features between binding-site and non-binding-site residues for these specific examples, and surface mapping of weighted contact number highlights local packing variations underlying functional distinctions. These case studies demonstrate how ProtDiS captures interpretable, feature-specific contributions at the residue level. **c**, Multi-level interpretability provided by ProtDiS. The heatmap shows knowledge-level SHAP attributions (green) across five rows: residual features (CK), secondary structure (SS), solvent accessibility (ASA), local packing (LP), and the sum of these four dimensions. The sixth row (red) corresponds to the true binding site annotation of each residue. The aggregated knowledge-level attribution broadly reflects residue-level functional relevance, highlighting interpretable structural determinants, though it does not perfectly coincide with the true binding sites.

complex geometric structures through the composition of strictly disentangled states, rather than indiscriminately allocating this information to the residual (Common Knowledge, CK) channel.

**Experimental Setup** To validate this mechanism, we evaluated the capacity of individual and combined decoupled channels to distinguish CDR loops from Framework Regions (FRs). We extracted 1,003 antibody structures from the SAbDab database and trained linear probes to classify residues as either CDR or FR based on the learned representations.

**Quantitative Results** As presented in Table 3, the empirical results demonstrate the effectiveness of the disentanglement process. The residual (CK) channel yields a relatively low ROC-AUC score of 0.6238. This confirms that the CK channel is effectively regularized and does not serve as an unconstrained information shortcut for complex geometries.

Conversely, the concatenation of all specific knowledge channels (*Combined Specific*) achieves a significantly higher ROC-AUC of 0.8235. When analyzing individual Specific Knowledge (SK) channels, Secondary Structure (SS) and Stability emerge as the most predictive features, scoring 0.7486 and 0.7050 respectively.

*Table 3.* ROC-AUC scores for distinguishing CDR loops from Framework Regions (FRs) using linear probes trained on 1,003 SAbDab antibodies. SK denotes Specific Knowledge channels.

| Feature Set Used for Linear Probe | ROC-AUC Score |
|---|---|
| Common Knowledge (CK / Residual) | 0.6238 |
| Single SK: Secondary Structure (SS) | 0.7486 |
| Single SK: Stability | 0.7050 |
| Single SK: Hydrophobicity (HYDRO) | 0.6536 |
| Single SK: B-factor | 0.6003 |
| Single SK: Local Packing (LP) | 0.5928 |
| Single SK: Solvent Accessibility (ASA) | 0.5769 |
| Single SK: Entropy | 0.5747 |
| Single SK: Orientational Coordinate (ORC) | 0.5687 |
| Combined Specific (All SKs concatenated) | 0.8235 |
| All Channels (CK + All SKs) | **0.8347** |

**Discussion** The high predictive power of the combined specific channels proves that structural variability is captured via the synergistic composition of decoupled biophysical features. Furthermore, the prominence of the SS and Stability channels aligns precisely with the underlying biology: CDRs typically manifest as flexible, unstructured loops with lower structural stability compared to the highly rigid and conserved beta-sheet architecture of the FRs.

## C.3. Analysis of Residual Channel Information Content

A potential concern regarding decoupled representation frameworks is whether the residual channel (Common Knowledge, CK) inadvertently becomes an unconstrained information bottleneck, particularly for poorly annotated folds, thereby bypassing the explicit disentangled pathways. It is important to note that the supervision for the explicit pathways in `ProtDiS` is derived directly from 3D atomic coordinates (e.g., via DSSP algorithms) rather than relying on database annotations. Consequently, orphan or novel folds consistently receive dense biophysical supervision. To further empirically validate the behavior of the CK channel, we conducted analyses across varying degrees of annotation sparsity and structural noise.

**Impact of Annotation Sparsity** We evaluated the representation behavior by partitioning CATH folds into two subsets: Highly Annotated groups ($\geq 50$ members) and Poorly Annotated groups ($\leq 3$ members or orphan folds). As illustrated in Figure 12, the variance of the CK channel remains remarkably stable across both subsets (0.5819 for Highly Annotated vs. 0.5839 for Poorly Annotated). Furthermore, the intrinsic dimensionality of the CK channel remains identical ($d = 7$) in both scenarios.

To confirm that the CK channel does not secretly encode bypassed physical information in poorly annotated folds, we performed linear probing specifically on the Poorly Annotated group. As shown in Figure 13, the Specific Knowledge (SK)

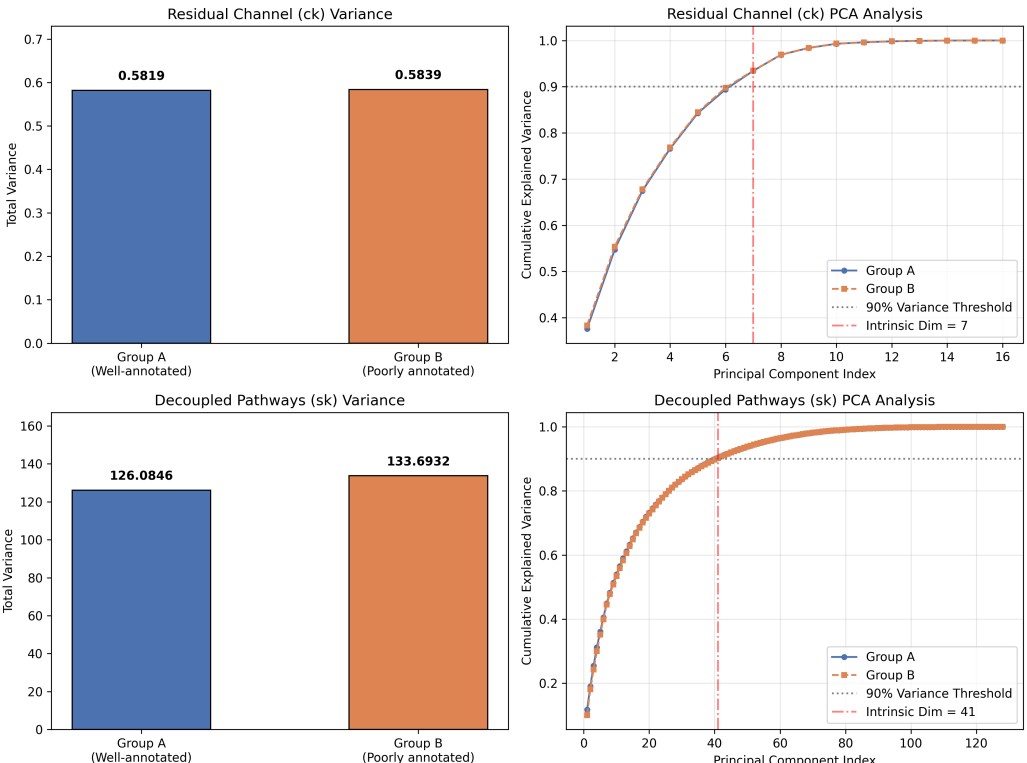

*Figure 12.* Comparison of the Common Knowledge (CK) channel variance and intrinsic dimensionality between Highly Annotated and Poorly Annotated CATH folds.

pathways maintain strong predictive performance for physical traits ($r > 0.8$), whereas the CK channel fails to predict these attributes ($r < 0.15$).

**Robustness to Structural Noise** In addition to annotation sparsity, we investigated whether low-quality structures force the model to dump unresolved information into the residual channel. We compared representations derived from High-Quality structures (resolution $\leq 2.5$ Å) against those from Low-Quality structures (resolution $\geq 4.0$ Å or low pLDDT scores).

As depicted in Figure 14, the variance of the CK channel remains tightly bounded, exhibiting only a marginal shift of +0.89% when processing low-quality inputs.

**Discussion** The stability of the CK variance and its inability to predict physical traits in challenging scenarios (orphan folds and noisy structures) confirm that the residual channel in `ProtDiS` strictly functions as a bounded, supplemental pathway. It does not act as a compensatory shortcut that bypasses the intended physical disentanglement.

### C.4. Systematic Ablation Studies in Feature Space

To rigorously evaluate the contribution of each algorithmic component without introducing confounding variables from downstream task-specific dataset biases or predictor head architectures, we conducted systematic ablation studies directly within the feature representation space.

**1. Necessity of Cross-Channel Decorrelation ($\mathcal{L}_{red}$)** We evaluated the impact of the decorrelation penalty using the inter-channel Distance Correlation (DCC). As shown in Figure 15, in the absence of $\mathcal{L}_{red}$, the decoupled channels suffer from severe non-linear feature collapse. For instance, the DCC between Local Packing and Hydrophobicity surges to 0.4688. Conversely, the full `ProtDiS` model consistently suppresses the inter-channel DCC to $0.08 \sim 0.09$. These results demonstrate that the decorrelation penalty effectively forces the sub-networks into non-redundant, orthogonal subspaces.

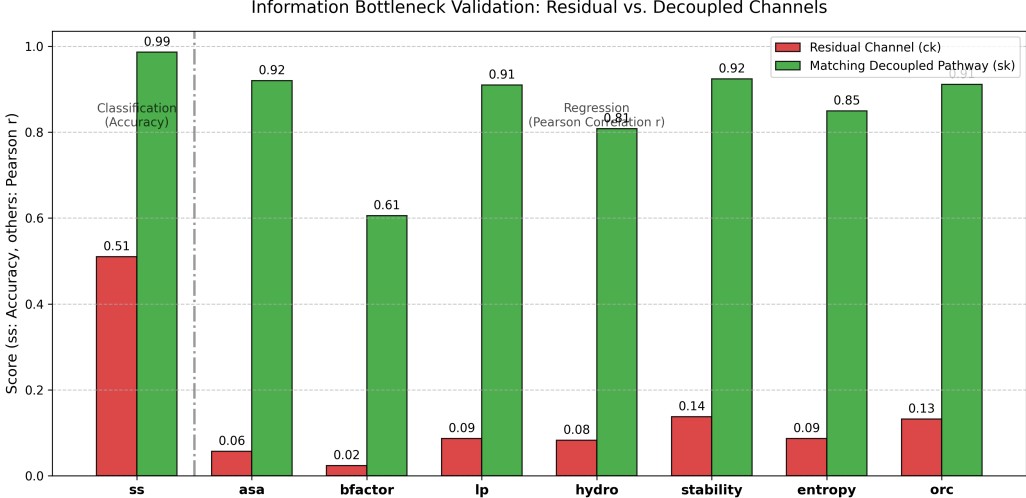

*Figure 13.* Linear probing performance (Pearson $r$) on Poorly Annotated folds, demonstrating that physical traits are correctly captured by SK pathways rather than the CK channel.

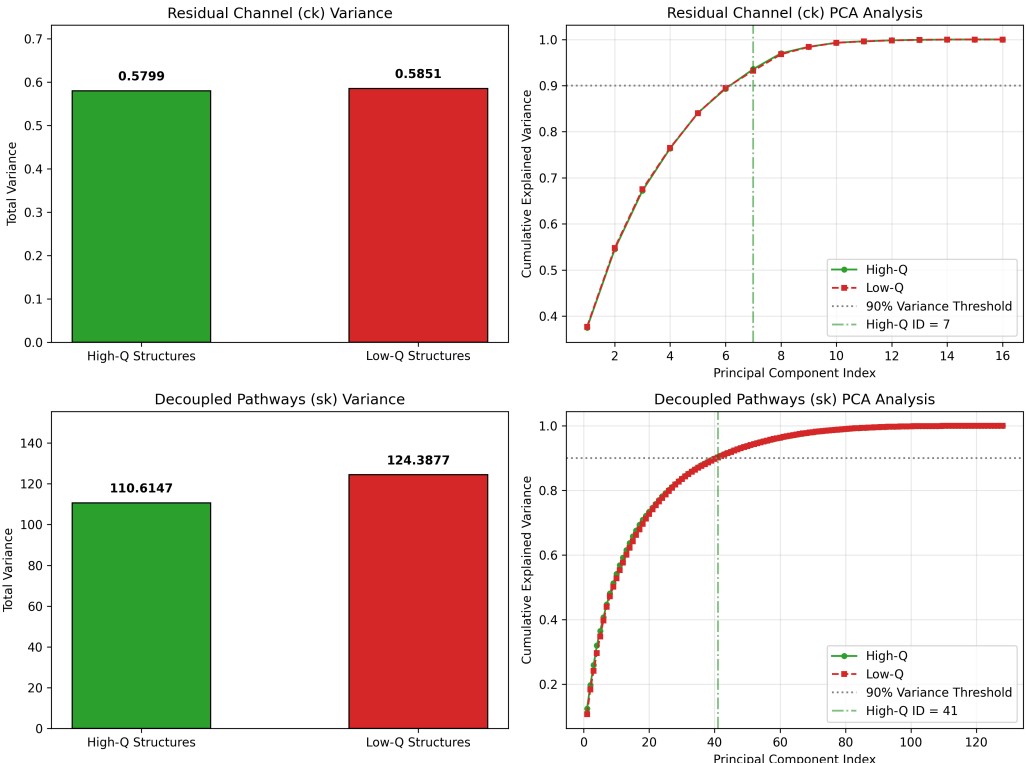

*Figure 14.* Distribution of CK channel variance shift when processing High-Quality versus Low-Quality protein structures.

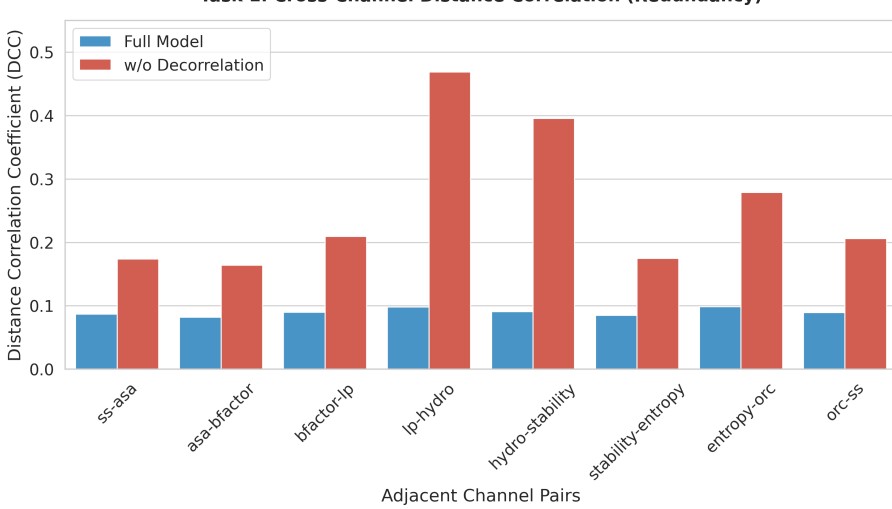

*Figure 15.* Inter-channel Distance Correlation (DCC) comparison between the ablated model without the decorrelation penalty and the full `ProtDiS` model.

**2. Necessity of Adversarial Removal ($\mathcal{L}_{adv}$)** The effectiveness of the adversarial removal mechanism was evaluated by training linear probes directly on the residual channel (Common Knowledge, CK). Without the adversarial loss $\mathcal{L}_{adv}$, the CK channel acts as an unconstrained reservoir, accurately predicting predefined attributes (e.g., ASA achieves 0.513 ($R^2$), Secondary Structure Accuracy = 68.6 (accuracy))). In the full model, however, this predictive power collapses to near-random guessing (ASA 0.005 ($R^2$)), as depicted in Figure 16. This empirical evidence indicates that the adversarial mechanism successfully purifies the residual channel, restricting it to unannotated structural semantics rather than bypassed specific knowledge.

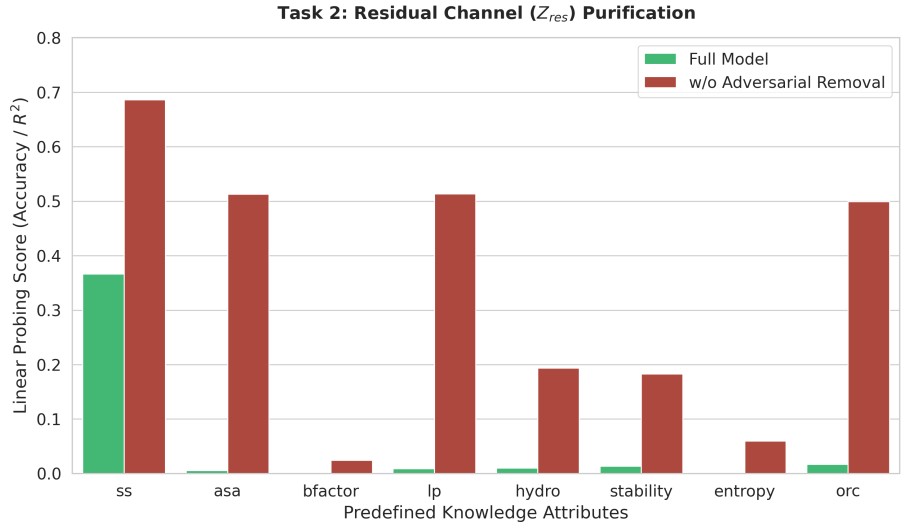

*Figure 16.* Predictive performance of linear probes trained on the residual (CK) channel with and without adversarial removal.

**3. Role of the KL Bottleneck ($\mathcal{L}_{KL}$)** The regularization effect of the KL divergence bottleneck was assessed using a non-linear MLP probe on the unseen test set. Removing the KL bottleneck $\mathcal{L}_{KL}$ allows the model to overfit by memorizing high-frequency structural noise. The introduction of $\mathcal{L}_{KL}$ acts as a crucial manifold regularizer, compelling the model to learn abstracted and smooth representations. Consequently, as illustrated in Figure 17, the full model achieves superior generalization, with performance on metrics such as contact entropy improving from 0.867 to 0.881.

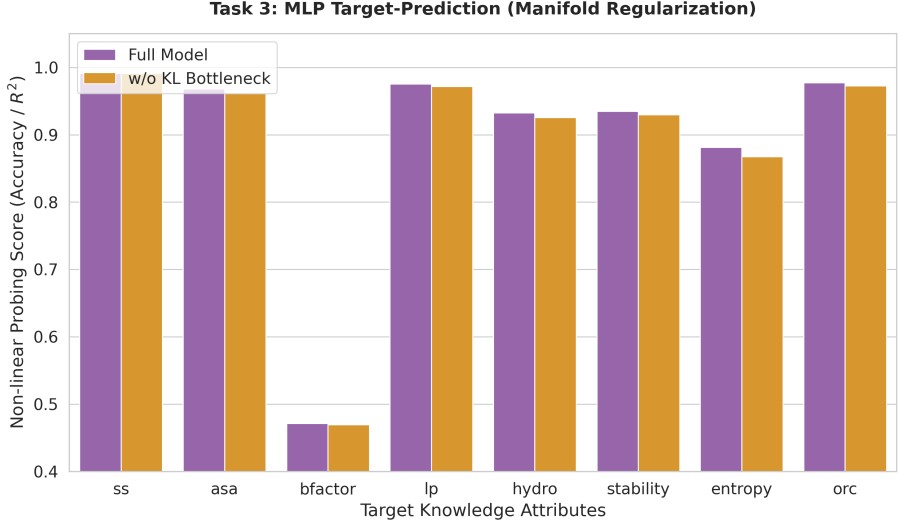

*Figure 17.* Generalization performance on the unseen test set evaluated via non-linear MLP probes, highlighting the regularization effect of the KL bottleneck.

