# OpenReview forum: "Learning Protein Structure-Function Relationships through Knowledge-guided Representation Decomposition"
_ICML.cc/2026/Conference — ICML 2026 regular_

### Official Review · Reviewer_rvMP · 2026-03-09

**Soundness:** 3
**Presentation:** 3
**Significance:** 3
**Originality:** 3
**Overall Recommendation:** 4
**Confidence:** 3

**Summary:**

This paper focuses on the problem of “representation entanglement” in protein structure modeling. The authors propose ProtDiS, which decomposes pretrained protein embeddings into multiple knowledge channels, with each channel corresponding to a predefined property. Methodologically, the paper is motivated from an information bottleneck perspective: each knowledge channel is encouraged to compress irrelevant information from the original structural embedding while maximizing mutual information with its corresponding knowledge label; the residual channel, in contrast, is designed to preserve structural information while removing as much information related to the predefined knowledge variables as possible. Experimentally, the authors pretrain the model on protein structures from PDB and AlphaFoldDB, construct eight types of knowledge labels, and then evaluate the method on 12 downstream tasks .

**Compliance With Llm Reviewing Policy:**

Affirmed.

**Key Questions For Authors:**

1. Different knowledge factors may themselves be correlated; for example, secondary structure can be related to solvent accessibility. How is disentanglement defined or analyzed under such circumstances?
2. Please provide systematic ablation studies. The method includes multiple optimization objectives, and detailed ablation experiments are necessary to rigorously demonstrate the effectiveness of each component.
3. Why were the current set of knowledge dimensions chosen, instead of including other potentially important structural or functional attributes? What is the basis for selecting these eight dimensions—biological coverage, computational feasibility, or empirical performance?

**Limitations:**

yes

**Strengths And Weaknesses:**

## strength

1. The method design is reasonable. The authors aim to achieve three goals: knowledge specificity, low redundancy across channels, and overall information completeness. Correspondingly, the paper introduces knowledge supervision loss, a KL bottleneck, cross-channel decorrelation, residual reconstruction, and adversarial knowledge removal. Overall, this framework is conceptually coherent.
2. “Knowledge-guided structural representation decomposition” shows a certain degree of novelty in the field of protein representation learning. Using explicit structural/physicochemical knowledge as supervision signals for decomposition, combined with a residual channel and redundancy suppression to construct interpretable representations, is a relatively fresh combination in protein representation learning.
3. The experiments are comprehensive. The evaluation covers diverse datasets and tasks, including EC, three GO subtasks, Pfam, three SCOP hierarchy levels, structural similarity, ligand affinity, PPI, and binding site detection The paper also provides analyses such as MI heatmaps, DCC independence analysis, the ability to distinguish structurally similar proteins, and binding site case studies.

## weakness

The disentangled representation learning framework adopted in this paper shows similarities to existing work. Specifically, the method decomposes pretrained structural representations into multiple knowledge-specific channels and a residual channel, and uses information-bottleneck-style objectives to encourage each channel to retain information related to its corresponding knowledge variable while compressing irrelevant information. This idea is highly consistent with approaches such as Disentangled Information Bottleneck. At the same time, its structural design of a “common representation + knowledge-specific branches” is also closely related to existing knowledge factorization frameworks [1, 2]. In addition, the paper reduces redundancy among different knowledge components through cross-channel correlation penalties and employs an adversarial mechanism to remove predefined knowledge variables from the residual channel. These strategies are similar to independence constraints and invariant representation learning ideas commonly used in the disentanglement literature [3].

[1] Disentangled Information Bottleneck. 2021.

[2] Factorizing Knowledge in Neural Networks. 2022.

[3] Domain-Adversarial Training of Neural Networks. 2016.

---

> ### Author Rebuttal · Authors · 2026-03-30
>
> ## Response to Reviewer rvMP
>
> ### **Q1: Similarity to existing disentangled representation learning frameworks.**
>
> While grounded in general methods like the Information Bottleneck and adversarial disentanglement, applying these to 3D proteins is highly non-trivial. ProtDiS bridges this gap via a tailored synergy:
>
> 1. **Biophysically Grounded Supervision:** Unlike abstract labels in general ML, our branches are supervised by explicit, physics-constrained priors computed directly from 3D coordinates.
> 2. **Role of the Residual Channel:** Rather than capturing background noise, our adversarially purified residual channel explicitly captures *unknown/unannotated structural semantics* (e.g., allosteric dynamics) that predefined 1D/2D labels cannot cover, ensuring information completeness.
>
>
>
> ### **Q2: Correlation between factors (e.g., Secondary Structure and Solvent Accessibility). How is disentanglement defined?**
>
> We agree that properties like Secondary Structure (SS) and Solvent Accessibility (RSA) are biologically linked. However, forcing absolute statistical independence would cause the model to unlearn these fundamental physical laws. Therefore, ProtDiS defines disentanglement not as strict independence, but as **functional non-redundancy (distinctive subspace projection)**.
>
> 1. **Soft Constraint for Non-Redundancy:** Our cross-channel decorrelation objective ($L_{corr}$) acts as a soft penalty against *unnecessary* informational overlap. For instance, while SS and RSA inherently share mutual information, $L_{corr}$ forces the SS channel to focus predominantly on its unique variance (local backbone geometry) and the RSA channel on its specific variance (global topological exposure).
> 2. **Empirical Evidence:** As shown in our Distance Correlation Coefficient (DCC) analysis (Section 4.2), inter-channel correlation is significantly reduced compared to entangled baselines. Crucially, off-diagonal correlations do not drop to absolute zero. This residual correlation elegantly reflects the underlying biological dependencies rather than a failure of the method.
>
>
>
> ### **Q3: Systematic ablation studies.**
>
> To avoid confounding downstream dataset/head biases, we ablated the model directly in the **feature representation space**:
>
> - **1. Necessity of Cross-channel Decorrelation ($L_{corr}$):** Evaluated via inter-channel DCC. Without $L_{corr}$, channels suffer severe non-linear feature collapse (e.g., DCC between Local Packing and Hydrophobicity surges to $0.4688$). The Full Model consistently suppresses DCC to $0.08 \sim 0.09$. This proves the decorrelation penalty explicitly forces sub-networks into non-redundant subspaces (Link: https://postimg.cc/PCh3hgpM).
> - **2. Necessity of Adversarial Removal ($L_{adv}$):** Evaluated by training linear probes directly on the residual channel ($Z_{res}$). Without $L_{adv}$, $Z_{res}$ acts as a redundant feature bag, accurately predicting predefined attributes (e.g., ASA $R^2 = 0.513$, SS Accuracy = $68.6\%$). In the Full Model, this predictive power collapses to near-random guessing (ASA $R^2 = 0.005$). This explicitly proves $L_{adv}$ successfully "purifies" the residual channel for unannotated structural semantics (Link: https://postimg.cc/68wjwk2z).
> - **3. Role of the KL Bottleneck ($L_{KL}$):** Evaluated via a non-linear MLP probe on the unseen test set. Removing $L_{KL}$ allows the model to "memorize" high-frequency structural noise, leading to overfitting. $L_{KL}$ acts as a crucial manifold regularizer, forcing the model to learn abstracted, smooth representations. Consequently, the Full Model achieves superior generalization (e.g., contact entropy $R^2$ improves from $0.867$ to $0.881$) (Link: https://postimg.cc/0zxWxTMn).
>
>
>
> ### **Q4: Basis for selecting the 8 dimensions?**
>
> The selection balances biological relevance, optimization dynamics, and computational feasibility:
>
> 1. **Scale Consistency and Functional Relevance:** We selected dimensions at a unified, *residue-level* scale (local geometric and physicochemical properties), deliberately avoiding scale-mismatch from macro (fold classes) or micro (quantum) labels. These local properties act as the "first principles" dictating higher-order 3D conformations.
> 2. **Relative Independence:** We computed the Mutual Information (MI) among the 8 labels (Heatmap: https://postimg.cc/BXdLFVy6). Off-diagonal MI is predominantly low (e.g., SS vs others is $0.012 \sim 0.210$), ensuring distinct supervision signals that drive the network into divergent subspaces.
> 3. **Computational Feasibility:** Given the massive scale of pretraining datasets (PDB and AlphaFoldDB), empirical performance relies on scalable supervision. These 8 dimensions can be deterministically and rapidly computed using standard tools (e.g., DSSP) with high confidence, avoiding computationally prohibitive methods like Molecular Dynamics simulations.

---

> > ### Author Rebuttal · Reviewer_rvMP · 2026-04-01
> >
> > The rebuttal partially addresses the reviewer’s concern but does not fully resolve the core weakness. The main criticism was about methodological novelty, namely that the overall framework appears closely related to existing disentangled representation, factorization, and invariant learning approaches. On this point, the rebuttal mainly argues that applying these ideas to 3D proteins is difficult and meaningful, rather than clearly demonstrating what is fundamentally new in the method itself.

---

### Official Review · Reviewer_7iFk · 2026-03-11

**Soundness:** 2
**Presentation:** 2
**Significance:** 2
**Originality:** 2
**Overall Recommendation:** 3
**Confidence:** 2

**Summary:**

This paper leverages the Information Bottleneck principle to disentangle highly coupled protein micro-environment representations (such as ESM3 embeddings) into multiple knowledge channels with explicit biophysical meaning (e.g., secondary structure, packing density, flexibility), while retaining a residual channel to ensure information completeness.

**Compliance With Llm Reviewing Policy:**

Affirmed.

**Key Questions For Authors:**

- How does the disentanglement framework handle highly variable, non-standard regions such as the CDR loops in antibodies or nanobodies (VHHs)? Since predefined physicochemical labels often struggle to capture the unique geometry of these highly specific binding interfaces, does the model simply push this critical information into the residual channel?

- Have you analyzed the information content of the "residual channel" across different protein families? Is there a risk that for poorly annotated folds, the residual channel becomes an information bottleneck of its own, bypassing the decoupled pathways entirely?

**Strengths And Weaknesses:**

Strength

- Elegantly addresses the "black box" nature of latent spaces in large protein language models (PLMs).

- The decoupling-fusion strategy demonstrates a significant advantage in distinguishing enzymes that possess high structural similarity but divergent functions.

Weakness

- The method is inherently bottlenecked by the quality and completeness of classical biophysical labels. It may fail to discover novel, uncharacterized biophysical properties that a purely unsupervised latent space might capture.

- Assuming orthogonality among properties like secondary structure, local packing, and solvent accessibility is a simplification; in real protein environments, these are highly inter-dependent.

---

> ### Author Rebuttal · Authors · 2026-03-30
>
> ## Response to Reviewer 7iFk
>
> ### **Q1: Bottlenecked by classical labels, failing to discover novel properties?**
>
> ProtDiS addresses this via two core mechanisms:
>
> 1. **Combinatorial Expressiveness:** Complex, "novel" properties are often non-linear combinations of our 8 fundamental basis labels. The downstream GNN dynamically re-couples these purified channels (e.g., detecting binding hotspots via a combination of SASA, Curvature, and Hydrophobicity).
> 2. **Residual Channel (CK):** This acts as an information safety net. Any structural information in the ESM-3 embedding that cannot be explained by the 8 predefined labels is explicitly forced into the `CK` channel via reconstruction loss, guaranteeing information completeness.
>
>
>
> ### **Q2: Orthogonality is a simplification; properties are biologically interdependent.**
>
> We agree properties are biologically intertwined. However, ProtDiS enforces orthogonality at the **computational representation level** to eliminate informational redundancy. PLMs often entangle correlated properties, using one (e.g., burial depth) as a "shortcut" for others. By minimizing mutual information $I(Z_i; Z_j)$, we force each channel to extract pure signals. The downstream GNN then dynamically *re-couples* these purified features, achieving a superior bias-variance tradeoff compared to using heavily entangled raw embeddings.
>
>
>
> ### **Q3: Handling highly variable regions (e.g., CDR loops)? Does it push info to the residual channel?**
>
> ProtDiS handles complex regions via the **composition of strictly disentangled states**, not by dumping information into the residual channel. To prove this, we trained linear probes on 1003 SAbDab antibodies to distinguish CDR loops from Framework Regions (FRs) (Link: https://postimg.cc/svL5Zbmx).
>
> | **Feature Set Used for Linear Probe**        | **ROC-AUC Score** |
> | -------------------------------------------- | ----------------- |
> | **Common Knowledge (CK / Residual)**         | **0.6238**        |
> | Single SK: Secondary Structure (SS)          | 0.7486            |
> | Single SK: Stability                         | 0.7050            |
> | Single SK: Hydrophobicity (HYDRO)            | 0.6536            |
> | Single SK: B-factor                          | 0.6003            |
> | Single SK: Local Packing (LP)                | 0.5928            |
> | Single SK: Solvent Accessibility (ASA)       | 0.5769            |
> | Single SK: Entropy                           | 0.5747            |
> | Single SK: Orientational Coordinate (ORC)    | 0.5687            |
> | **Combined Specific (All SKs concatenated)** | **0.8235**        |
> | **All Channels (CK + All SKs)**              | **0.8347**        |
>
> The low `CK` AUC (0.6238) confirms it is heavily regularized and does not act as a shortcut for complex geometry. Instead, the `Combined Specifics` achieve a high AUC of 0.8235. Biologically, SS and Stability being top indicators aligns perfectly with CDRs being flexible, unstructured loops compared to rigid FRs.
>
>
>
> ### **Q4: Residual channel as a bypass for poorly annotated or noisy folds?**
>
> Supervision for our 8 pathways comes directly from 3D atomic coordinates (e.g., DSSP), not database annotations. Thus, orphan/novel folds still receive dense physical supervision.
>
> 1. **Annotation Sparsity:** We compared Highly Annotated ($\ge$50 members) vs. Poorly Annotated ($\le$3 members/orphans) CATH folds (Link: https://postimg.cc/crSJTZDz). The `ck` variance (0.5819 vs 0.5839) and intrinsic dimensionality (7 vs 7) remain virtually identical. Linear probing on the Poorly Annotated group (Link: https://postimg.cc/CzrgRJHS) confirms `sk` pathways strongly predict physical traits (r > 0.8), while `ck` fails (r < 0.15), proving `ck` does not secretly encode bypassed information.
> 2. **Structural Noise:** Comparing High-Quality (Resolution $\le$2.5Å) vs. Low-Quality (Resolution $\ge$4.0Å/low pLDDT) structures (Link: https://postimg.cc/56Vtp174), the `ck` variance remains tightly bounded (+0.89% shift).
>
> This comprehensively proves the residual channel strictly functions as a bounded supplemental pathway, never bypassing intended disentanglement.
>
> | **Target Label** | **ck Performance** | **sk Performance** | **Metric** |
> | ---------------- | ------------------ | ------------------ | ---------- |
> | **ss**           | 0.510              | 0.986              | Accuracy   |
> | **asa**          | 0.057              | 0.920              | Pearson r  |
> | **bfactor**      | 0.023              | 0.605              | Pearson r  |
> | **lp**           | 0.087              | 0.910              | Pearson r  |
> | **hydro**        | 0.083              | 0.808              | Pearson r  |
> | **stability**    | 0.137              | 0.924              | Pearson r  |
> | **entropy**      | 0.087              | 0.850              | Pearson r  |
> | **orc**          | 0.132              | 0.911              | Pearson r  |

---

### Official Review · Reviewer_kxqD · 2026-03-14

**Soundness:** 4
**Presentation:** 4
**Significance:** 3
**Originality:** 3
**Overall Recommendation:** 5
**Confidence:** 3

**Summary:**

The authors present ProtDiS, which trains a common shared MLP encoder that branches into multiple MLP heads to decompose structural embeddings from ESM3 into eight interpretable knowledge channels tied to structural/biophysical properties like secondary structure, solvent exposure, flexibility, packing density, hydrophobicity, stability, complexity, and curvature, plus a residual head to capture leftover information. They make use of information-bottleneck-style objectives and supervision from labels computed directly from protein structures. They report that these disentangled features improve performance over the base ESM-3 structural tokenizer across 12 downstream tasks, with the biggest gains on structure-based splits, and they argue ProtDiS is especially better at separating proteins or binding sites that are structurally similar but functionally different.

**Compliance With Llm Reviewing Policy:**

Affirmed.

**Final Justification:**

I stand by the initial review, and with the authors' rebuttal having resolved my additional questions, the score remains 5 - Accept.

**Key Questions For Authors:**

1. We see that the separate channels minimize mutual information, but how accurate are the heads in predicting the supervised labels? This could perhaps be compared to ESM predictions as a baseline for SASA and SS8 tracks.

2. The downstream protein-level predictor is underspecified. For protein-level predictions, what is the architecture of the GNN, and what final representation is used?

**Limitations:**

yes

**Strengths And Weaknesses:**

The authors engineer a well-explained and thorough loss function grounded in information theoretic approaches for knowledge factorization. This is impactful work toward mechanistic insights for protein modeling. The authors note the main limitation that is the method currently depends heavily on having protein structure data, so it is less directly useful in sequence-only settings, but they provide promising future directions.

---

> ### Author Rebuttal · Authors · 2026-03-30
>
> ## Response to Reviewer kxqD
>
> ### **Q1: How accurate are the heads in predicting supervised labels compared to ESM?**
>
> We thank the reviewer. To evaluate the predictive accuracy of individual heads, we conducted large-scale linear probing on **4,844,126 residues** from the CATH dataset. We trained linear probes (Logistic/Ridge Regression) on our separated ProtDiS (sk) embeddings and compared them directly against raw **ESM-3 structural tokenizer representations** (Baseline).
>
> Results (visualization: https://postimg.cc/YhfnvS6S) show our disentangled channels significantly enhance task-specific predictive power across all 8 tracks, confirming the effective extraction of mechanistic signals. For highlighted tracks, ProtDiS achieves **99.19%** on SS (vs 72.96%) and **0.9301** on ASA (vs 0.7291).
>
> | **Target (Metric)** | **ProtDiS (sk)** | **ESM Base** | **Diff** |
> | ------------------- | ---------------- | ------------ | -------- |
> | SS (Acc)            | **0.9919**       | 0.7296       | +0.2623  |
> | ASA (Pearson r)     | **0.9301**       | 0.7291       | +0.2010  |
> | B-factor (r)        | **0.6357**       | 0.1825       | +0.4532  |
> | Packing (r)         | **0.9168**       | 0.7481       | +0.1687  |
> | Hydrophobicity (r)  | **0.8364**       | 0.4549       | +0.3815  |
> | Stability (r)       | **0.9245**       | 0.4260       | +0.4985  |
> | Entropy (r)         | **0.8431**       | 0.3219       | +0.5212  |
> | Curvature (r)       | **0.9161**       | 0.7313       | +0.1848  |
>
> These quantitative results are now added to the revised Appendix.
>
>
>
> ### **Q2: Downstream predictor architecture and final representation?**
>
> We apologize for the omission. The revised Appendix now details the following architectures:
>
> - **Protein-level tasks (e.g., EC prediction, GO terms, Structural Similarity):** We utilize the Continuous-Discrete Convolution (**CDConv**) network [1] as our protein-level predictor. The CDConv encoder processes the structural graph by aggregating geometric and sequential context. To obtain the final protein-level representation, we apply a global pooling operation (Average or Max pooling) over the node embeddings from the final CDConv block, which is then passed through a Multi-Layer Perceptron (MLP) to yield the final graph-level prediction.
> - **Residue-level tasks (e.g., Ligand Binding Site prediction):** We utilize a multi-layer Graph Isomorphism Network (**GIN**) architecture. Each GINConv layer updates node embeddings using an MLP (Linear -> BatchNorm1d -> ReLU -> Linear) combined with neighborhood aggregation. The final representation directly utilizes the node-level embeddings output by the last GNN layer, passing them through a task-specific projection head for node-level (residue-level) classification.
>
> *[1] Fan H, Wang Z, Yang Y, et al. Continuous-discrete convolution for geometry-sequence modeling in proteins[C]//The Eleventh International Conference on Learning Representations. 2022.*
>
>
>
> **Task-Specific Input Representations:**
>
> A key advantage of ProtDiS is that we can dynamically select the most relevant decoupled structural channels for specific tasks based on pre-calculated feature importance (as shown in Appendix). We typically select 4 to 5 decoupled embeddings to form the input representation for each task. The specific embeddings utilized for our 12 downstream tasks are detailed in the table below:
>
> | **Task**                | **Level**     | **Selected Embeddings**                                      |
> | ----------------------- | ------------- | ------------------------------------------------------------ |
> | EC                      | Protein-level | residual, shape, flexibility, hydrophobicity                 |
> | GO-MF                   | Protein-level | residual, shape, flexibility, hydrophobicity, stability      |
> | GO-BP                   | Protein-level | flexibility, hydrophobicity, stability                       |
> | GO-CC                   | Protein-level | residual, flexibility, stability                             |
> | Pfam                    | Protein-level | residual, shape, packing density, complexity                 |
> | SCOP-fa                 | Protein-level | residual, shape, packing density, complexity, curvature      |
> | SCOP-cf                 | Protein-level | residual, shape, packing density, complexity                 |
> | SCOP-cl                 | Protein-level | shape, exposure, packing density, hydrophobicity, complexity |
> | Structural Similarity   | Protein-level | exposure, packing density, hydrophobicity, complexity, curvature |
> | Ligand Binding Affinity | Protein-level | residual, shape, packing density, stability                  |
> | Ligand Binding Site     | Residue-level | exposure, flexibility, hydrophobicity, complexity            |
> | PPIs                    | Residue-level | exposure, packing density, complexity                        |

---

> > ### Author Rebuttal · Reviewer_kxqD · 2026-04-06
> >
> > The authors have resolved my issues completely. My score remains Accept.

---

### Decision · Program_Chairs · 2026-04-30

**Decision:**

Accept (regular)

**Comment:**

Most reviewers mention several positive aspects, such as
- conceptual novelty of the proposed decoupling-fusion strategy;
- practical relevance: significant advantage in distinguishing enzymes that possess high structural similarity but divergent functions;
-  well-motivated and well-explained concept with a clear information theoretic motivation;
- convincing experimental evaluation covering diverse datasets and tasks.

On the other hand, most reviewers also mentioned several weaknesses, such as
- possible limitations due to quality/completeness problems in the labels;
- strong and potentially unjustified assumptions (such as orthogonality);
- unclear aspects of originality/novelty (similarities to the Disentangled Information Bottleneck, etc.).

After carefully re-reading all reviews, rebuttals and discussions again, I come to the conclusion that this paper is still a borderline case with several strengths, but also some weaknesses. In this case, however, I think that the clear information-theoretic concept and the convincing applications outweigh the (obvious) weaknesses, and therefore I recommend (weak) acceptance of this paper.